# Wnt4 and ephrinB2 instruct apical constriction via Dishevelled and non-canonical signaling

Jaeho Yoon [1] ✉, Jian Sun[1], Moonsup Lee[1], Yoo-Seok Hwang[1] & Ira O. Daar [1] ✉

Apical constriction is a cell shape change critical to vertebrate neural tube closure, and the contractile force required for this process is generated by actin-myosin networks. The signaling cue that instructs this process has remained elusive. Here, we identify Wnt4 and the transmembrane ephrinB2 protein as playing an instructive role in neural tube closure as members of a signaling complex we termed WERDS (Wnt4, EphrinB2, Ror2, Dishevelled (Dsh2), and Shroom3). Disruption of function or interaction among members of the WERDS complex results in defects of apical constriction and neural tube closure. The mechanism of action involves an interaction of ephrinB2 with the Dsh2 scaffold protein that enhances the formation of the WERDS complex, which in turn, activates Rho-associated kinase to induce apical constriction. Moreover, the ephrinB2/Dsh2 interaction promotes non-canonical Wnt signaling and shows how cross-talk between two major signal transduction pathways, Eph/ephrin and Wnt, coordinate morphogenesis of the neural tube.

Neurulation is the process of forming a neural tube from a neural plate. The neural plate is derived from the neuroectoderm of blastula embryos[1]. In the neurula embryo, both lateral sides of the neural plate fold toward the midline of the embryo and make a neural groove, and then eventually the two edges merge to make the neural tube. This process is essential for embryonic development of the brain, spinal cord, and neural crest and is well conserved in fish, amphibians, birds, and mammals[2].

During neural tube closure, the neuroepithelial sheet undergoes two major morphogenetic tissue movements[3]: 1) Anteroposterior (AP) axis elongation by polarized cell division and mediolateral cell intercalation; 2) Neural plate folding by apical constriction. Various studies have demonstrated that several cellular signaling and gene networks participate in these cell behaviors[4]. Non-canonical Wnt signaling is necessary for proper AP axis elongation through oriented cell division[5]. Planar cell polarity (PCP) signaling also has been proposed to orchestrate the mediolateral cell intercalation[3,6,7]. Asymmetrically localized PCP core components modulate the polarization of cells via the cell contact-based interaction of PCP protein complexes[8]. For example, Flamingo (Fmi/Celsr)-Frizzled receptor (Fz)- Dishevelled

(Dsh/Dvl) are found at the posterior edge of the cells, while Fmi- Prickle (Pk)- Van Gogh-like 2 (Vangl2) are positioned at the anterior edges of cells[9]. These asymmetrically localized PCP proteins play a critical role in cell intercalation driven by Rho-Rock mediated actomyosin contractility at shrinking cell junctions[3]. In the vertebrate embryo, non-canonical Wnt ligands, such as Wnt4, 5, or 11, have been suggested to instruct the polarized localization of PCP components during neural tube closure, hair cell development in the mouse cochlea, and mouse limb development[10–14]. In contrast, recent studies examining *Drosophila* wing development demonstrated that Wnt ligands are dispensable for establishing PCP signaling[15,16]. In addition to signaling networks, mechanical forces have also been strongly implicated in the modulation of PCP during tissue morphogenesis[17–19].

Apical constriction is a polarized cell shape change along the apical-basal axis inducing tissue remodeling in many developmental processes such as neural tube closure[3]. Contraction of the actin filament (F-actin) and the molecular motor non-muscle myosin II (myosin) are key components driving apical constriction[20–24]. Shroom3 and Rock accumulate on the apical side of the epithelial cell layer during neural tube closure and generate mechanical forces by actomyosin

[1]Cancer & Developmental Biology Laboratory, Center for Cancer Research, National Cancer Institute, National Institutes of Health, Frederick, MD 21702, USA. ✉e-mail: jaeho.yoon@nih.gov; daari@mail.nih.gov

contraction[20,25,26]. Loss of Shroom3 induces neural tube closure defects in *Xenopus* embryos[27]. However, the molecular mechanism of instructing apical constriction at the appropriate position and the right time is poorly understood.

EphrinB1 and ephrinB2 have been reported to be required for proper neural tube closure[27,28]. EphrinB1 was shown to play a role in murine neural tube morphogenesis by promoting adhesion of apical progenitors[28], while the loss of ephrinB2 caused apical constriction defects in *Xenopus* resulting in incomplete neural tube closure[29], but the molecular mechanism is unknown. To gain mechanistic insight into the role of ephrinB2 during apical constriction, we employed the *Xenopus* embryo as a model system. The *Xenopus* embryo has certain advantageous characteristics such as a large size that allows facile surgical manipulation and the embryos grow externally making them very amenable to live-cell imaging.

In this work, we have identified a protein complex consisting of Wnt4, ephrinB2, Ror2, Dsh2, and Shroom3 (WERDS) that plays a substantive role in coordinating apical constriction during neural tube closure. Loss-of-function and gain-of-function analyses demonstrate that the WERDS signaling complex regulates contractile actin-myosin network formation during apical constriction. Furthermore, we provide evidence that ephrinB2 modulates the conformational change of Dsh2 to facilitate formation of the WERDS signaling complex. Moreover, an ephrinB2/Dsh2 interaction can antagonize Wnt/β-catenin signaling, supporting the role for WERDS in non-canonical Wnt signaling required for proper neural tube formation.

## Results

### EphrinB2 interacts with Ror2 during neural tube closure

EphrinB2 is a ligand for Eph receptors and regulates a variety of embryonic developmental processes[30–33]. Unlike Eph receptors, the ephrinB2 transmembrane ligand has a short intracellular domain[34] that allows it to transmit signals in its host cell through interacting with binding partners[35]. A previous study reported on the regulation of ephrinB2 protein processing during neural tube closure, but also indicated that ephrinB2 played a consequential role in apical constriction during this morphogenetic event[29]. To understand the molecular mechanism of how ephrinB2 regulates apical constriction, we performed a screen of ephrinB2-binding partners using IP/Mass spectrometry analysis from the neural plate explants of ephrinB2-V5 overexpressing *Xenopus* embryos (Fig. 1a). Among the candidates from the mass spectrometry results, we identified several candidate proteins from the non-canonical Wnt pathway, including Ror2, Dsh2, Wnt4, and Shroom3 (Supplementary Data 1). Several studies have demonstrated that Ror2 is involved in neural tube closure[36], and a double knockout of Ror2 with the PCP protein Vangl2 causes more severe neural tube closure defects than knockout of Vangl2 alone[37]. Dsh is also known to regulate neural tube closure as a key component of Wnt signaling[36], and has been shown to be mutated in human neural tube defects (NTDs)[36,38]. Overexpression of Wnt4 causes reorientation of polarity in hairs of fly wings, suggesting that Wnt4 regulates planar cell polarity (PCP)[10]. Knockdown of Wnt4 causes severe gastrulation defects during early *Xenopus* embryogenesis[39]. However, the molecular mechanism of how Ror2 or Wnt4 affect neural tube closure is not clearly understood. Another candidate from the IP analysis, Shroom3, is known to regulate apical constriction through ROCK activation[22]. Knockdown of Shroom3 displays NTDs in various animal models[20] and pathogenic mutations causes NTDs in humans[40].

To assess whether these candidates may have the opportunity to interact with ephrinB2 during neural tube formation, the spatial expression patterns were analyzed using hybridization chain reaction RNA fluorescence in situ hybridization (HCR RNA-FISH). As shown in Fig. 1b, ephrinB2 and Ror2 were expressed along the neural plate and notochord (NC). Wnt4 was expressed along the neural plate but excluded from the hinge region. Shroom3 was also expressed along

the neural plate. However, EphB4 (an ephrinB2 cognate receptor) was expressed in the presomitic mesoderm (PM) and the notochord (NC), while Wnt5a (a known ligand for Ror2) was expressed in PM, suggesting that these two proteins may be less likely to be involved in neural tube closure. These results raise the possibility that ephrinB2, Wnt4, Ror2, and Shroom3 may be involved in apical constriction during neural tube closure.

To validate whether an interaction between ephrinB2 and Ror2 is likely, we performed co-immunoprecipitation (Co-IP) analysis from embryos overexpressing tagged forms of the proteins. Of the three ephrin-B ligands, ephrinB2 has the most robust interaction with Ror2 (Fig. 1c). To determine the region within ephrinB2 that is necessary for the interaction with Ror2, deletion mutants in ephrinB2 were constructed (Supplementary Fig. 1a). Co-IP with the ephrinB2 mutants showed that the intracellular domain of ephrinB2 is required for the Ror2/ephrinB2 interaction (Supplementary Fig. 1b). Co-IP using serial deletion mutants of ephrinB2 determined that the C-terminal six amino acids, known as a PDZ-binding motif, are required for the ephrinB2/Ror2 interaction (Fig. 1d, f). To identify the region within Ror2 necessary for the interaction with ephrinB2, deletion mutants of conserved domains in Ror2 were generated (Supplementary Fig. 1c). Co-IP with these Ror2 deletion mutants showed that the C-terminal tail region is necessary for the interaction with ephrinB2 (Supplementary Fig. 1d), and more specifically, that the Proline-rich domain (PR) is required for the interaction (Fig. 1e, g). These data indicate that the ephrinB2/Ror2 interaction is dependent upon the ephrinB2 C-terminal six amino acids and the PR domain within Ror2.

Next, we asked whether the ephrinB2/Ror2 interaction is required for proper neural tube closure. Morpholino antisense oligonucleotides (MOs) against ephrinB2 (ephrinB2 MO) or Ror2 (Ror2 MO) were injected into the D.1.1 blastomere. These unilateral injections were performed such that the uninjected side served as an internal control. The D1.1 blastomere targeted injection allowed for the avoidance of broader loss of function effects that may cause compounding morphological phenotypes, such as gastrulation defects. Consistent with the previous report[29], MO-mediated ephrinB2 knockdown caused neural tube closure defects (Supplementary Movie 1), while injection of MO-resistant mRNA to re-express wildtype (WT) ephrinB2 rescued neural tube closure (Fig. 1h). However, expression of an equivalent amount of the ephrinB2-Δ4 mutant lacking the ability to interact with Ror2, failed to rescue the defect (Fig. 1h). In a complementary experiment, Ror2 MO caused neural tube closure defects (Supplementary Movie 1), similar with an ephrinB2 knockdown, and re-introducing WT-Ror2 rescued the defect (Fig. 1i). Interestingly, expressing Ror2- ΔFr (Wnt ligand binding mutant) or Ror2- ΔPR (ephrinB2-binding mutant) failed to rescue the Ror2 MO-mediated defect. These results suggest that an ephrinB2/Ror2 interaction is required for proper neural tube closure.

To exclude possible indirect causes for the aberrant neural tube closure, we examined whether ephrinB2 or Ror2 knockdown affects the fate specification of the neural plate and neural plate border. In situ hybridization analysis with anti *Sox2* (neural plate marker) or *Twist* (neural crest marker) showed that loss of ephrinB2 or Ror2 did not affect fate specification (Supplementary Fig. 1e). To further validate the loss-of-function phenotype conferred by the ephrinB2 MO or Ror2 MO, we employed a knockout strategy using CRISPR/Cas9. EphrinB2 or Ror2 crispants also showed neural tube closure defects confirming that ephrinB2 and Ror2 regulate this process (Supplementary Fig. 1f and Supplementary Movie. 2).

### EphrinB2 and Ror2 regulate contractile actin bundle formation

Having established that an interaction between ephrinB2 and Ror2 is required for neural tube closure, we asked whether apical constriction, a driving force for the neural plate folding, is similarly affected. In transverse sections through the neural plate, the cell boundaries were

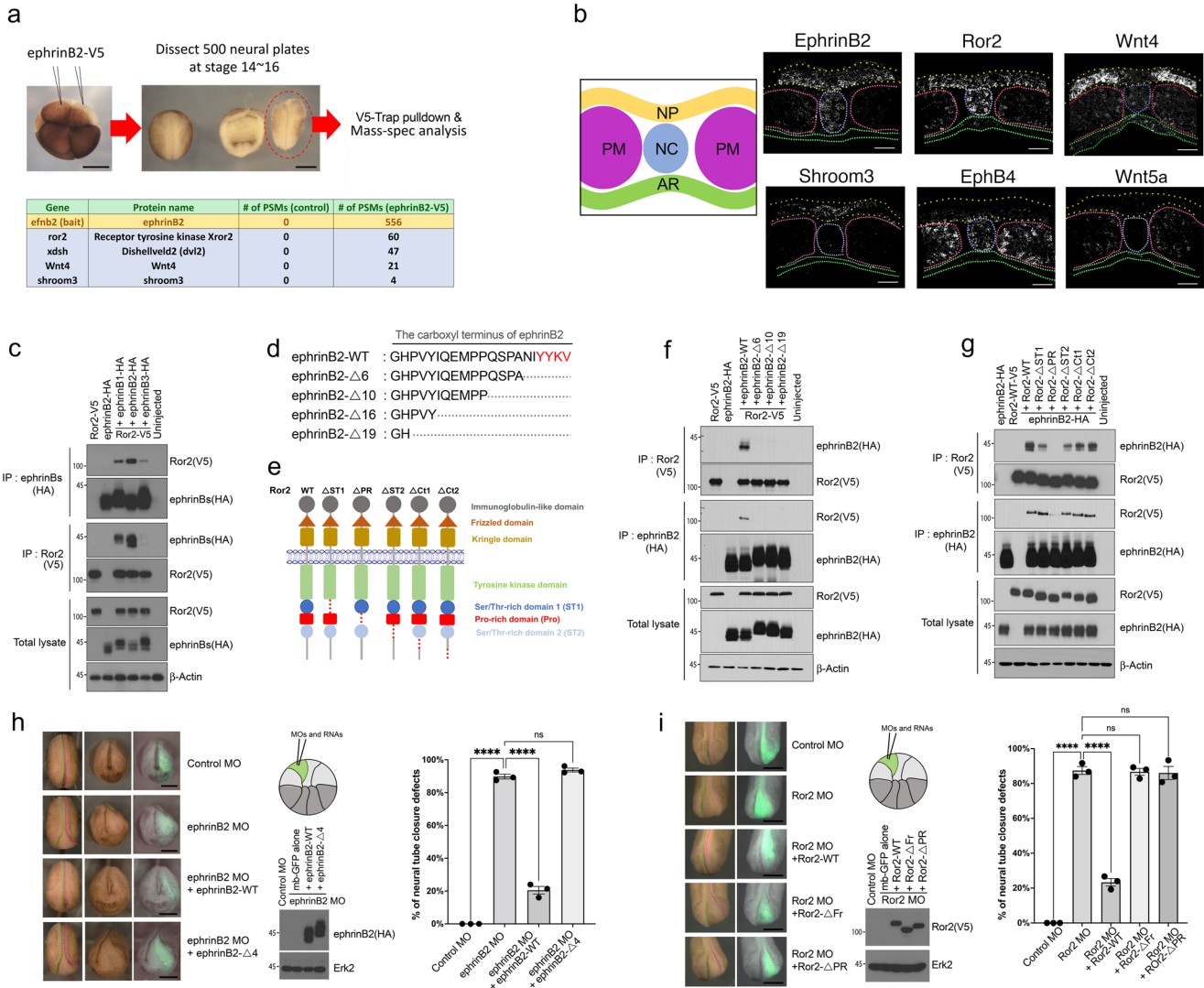

**Fig. 1 | ephrinB2/Ror2 interaction regulates apical constriction. a** Schematic of screening for ephrinB2-binding partners and selected candidates from ephrinB2 pulldown assay. Scale bar, 500 μm. See also Supplementary Data 1. **b** Schematic of the transverse section through neural plate and RNA in situ hybridization (HCR) assay. Neural plate (yellow), PM; pre-somatic mesoderm (magenta), NC; notochord (blue), AR; archenteron roof (green). Scale bar, 50 μm. **c** Co-IP assay with Ror2 (1 ng) and ephrin-B type ligands (500 pg) at Stage 13 *Xenopus* embryos. **d** Schematic of ephrinB2 serial deletion mutants in C-terminus. The PDZ-binding motif in red. **e** Schematic of Ror2 serial deletion mutants in the tail region. **f** Co-IP with Ror2 (1 ng) and ephrinB2 (500 pg) serial deletion mutants. **g** Co-IP with ephrinB2 (500 pg) and Ror2 serial deletion mutants (1 ng). **h** Representative embryo morphology by ephrinB2 knockdown (2 ng of MO). See also Supplementary Movie 1. Schematic of D1.1 microinjection in the 16-cell stage embryo. Membrane-GFP indicates injected side (50 pg). The lateral boundaries of the neural plate are marked by dotted lines (green; uninjected side and magenta; injected side).

Immunoblotting represents the expression level of ephrinB2 wildtype (WT) and Δ4 (25 pg). Data means SEM. Numbers of embryos/experiments: 85/3, 80/3, 78/3, and 79/3 from left to right for each column. One-way ANOVAs followed by Dunnett's multiple comparison tests were used ($p$-value <0.0001). n.s., not significant, ****$p$ < 0.0001. Scale bar, 500 μm. **i** Representative embryo morphology by Ror2 knockdown (2 ng of MO). See also Supplementary Movie 1. Schematic of D1.1 microinjection in the 16-cell stage embryo. Membrane-GFP indicates injected side. The lateral boundaries of the neural plate are marked by dotted lines (green; uninjected side and magenta; injected side). Immunoblotting represents expression levels of Ror2-WT, ΔFr, and ΔPR (50 pg). Data means SEM. Numbers of embryos/experiments: 79/3, 79/3, 81/3, 82/3, and 81/3 from left to right for each column. One-way ANOVAs followed by Dunnett's multiple comparison tests were used ($p$-value <0.0001). n.s., not significant, ****$p$ < 0.0001. Scale bar, 500 μm.. Scale bar, 500 μm.

labeled with membrane-RFP, while MOs and GFP mRNA were injected into one side of the neural plate to examine cell shape and apical constriction (Fig. 2a). Control MO-injected neural plate cells displayed a normal neural groove and narrow apical width (<10 μm). EphrinB2 or Ror2 knockdown in neural plate cells resulted in a significant expansion of the apical width (10–45 μm) (Fig. 2a). This apical constriction defect was rescued by re-expressing the wild-type counterparts of these target proteins. However, the binding mutants (ephrinB2-Δ4, Ror2-ΔFr or, Ror2-ΔPR) failed to rescue the defects. The results clearly indicate that ephrinB2 and Ror2 are involved in the apical constriction process during neural tube closure.

The actin-myosin network is a driving force for apical constriction[22]. To gain mechanistic insight into how the loss of ephrinB2 or Ror2 causes apical constriction defects, we analyzed the level of apical actin filaments. MO-mediated knockdown of ephrinB2 or Ror2 dramatically decreased apical F-actin levels as evidenced by fluorescence intensity of phalloidin staining in neural plate cells (Fig. 2b). Of particular note, re-expressing ephrinB2-WT or Ror2-WT in the morphant embryos rescued the apical F-actin levels, but expressing mutants that interfere with an interaction between ephrinB2 and Ror2 (ephrinB2-Δ4 or Ror2-ΔPR) or Wnt ligand binding to Ror2 (Ror2-ΔFr) showed no rescue of apical F-actin levels (Fig. 2b). Our results suggest that ephrinB2 and Ror2

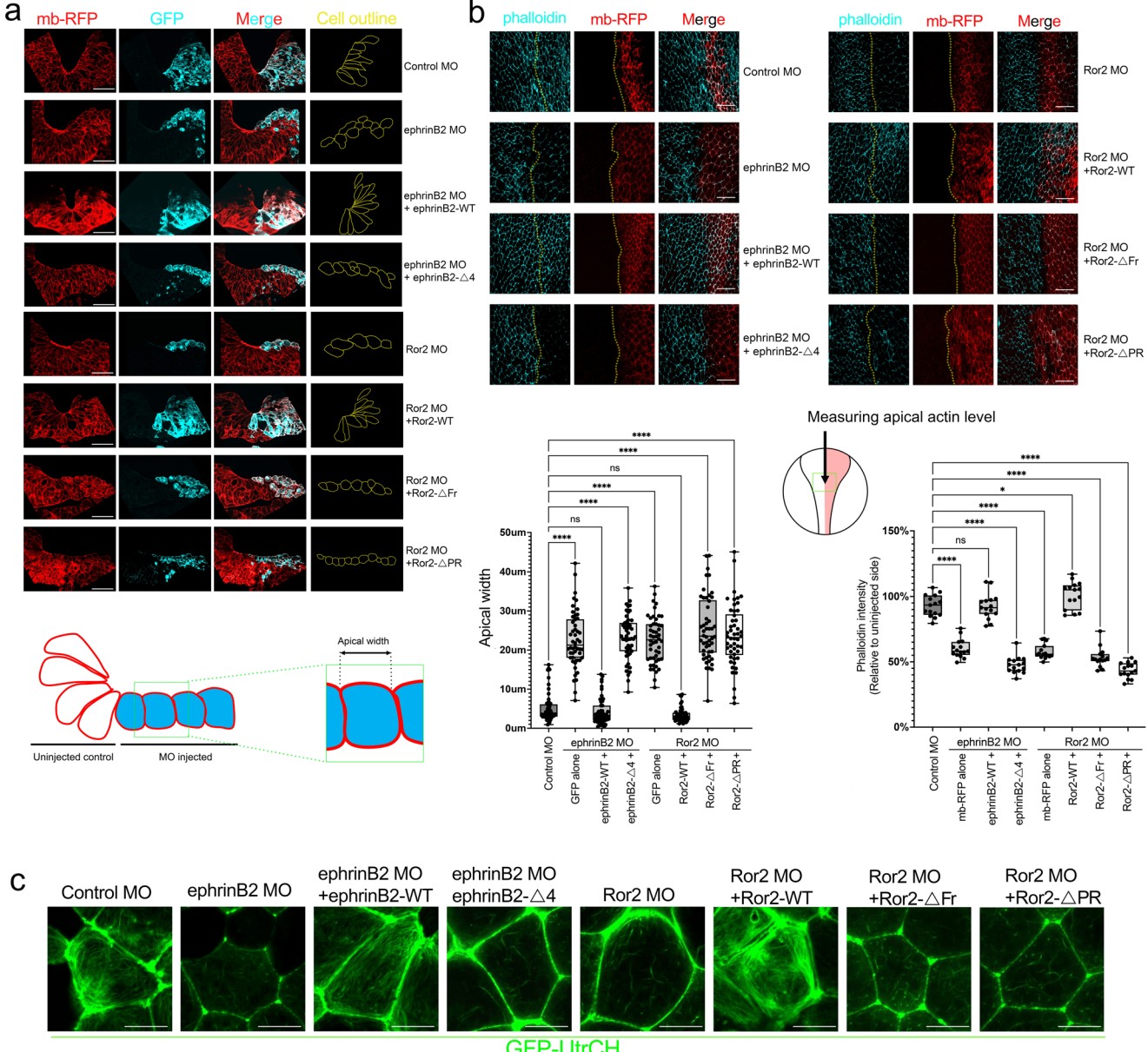

**Fig. 2 | ephrinB2/Ror2 interaction regulates contractile actin network formation. a** Transverse section through the neural plate. Unilateral injection of MO (2 ng) and RNA (ephrinB2s; 25 pg, Ror2s; 50 pg) as indicated. Membrane-RFP (100 pg) labeled outline of the neural plate cells. GFP (50 pg) indicates the MO-injected side. The box plot shows Min to Max with all data points. Numbers of embryos/experiments: 50/5 for each column. One-way ANOVAs followed by Dunnett's multiple comparison tests were used (*p*-value < 0.0001). n.s., not significant, ****p* < 0.0001. Scale bar, 500 μm. Schematic of measuring apical width. Scale bar, 50 μm. **b** Apical F-actin levels in neural plate cells by Phalloidin staining. Membrane-RFP indicates the injected side. The box plot shows Min to Max with all data points. Numbers of embryos/experiments: 15/3 for each column. One-way ANOVAs followed by Dunnett's multiple comparison tests were used. n.s., not significant, ****p* < 0.005, and *****p* < 0.0001. Schematic of area where images taken. Scale bar, 50 μm. **c** Stills from time-lapse imaging of GFP-UtrCH (100 pg) overexpressed neural plates from stage 14 to 15 embryos. See also Supplementary Movie 3 and 4. Scale bar, 10 μm.

regulate apical F-actin bundle formation during apical constriction. To examine the dynamic contractile actin bundle formation, we performed live-cell imaging using GFP-UtrCH. UtrCH is the calponin homology domain of utrophin, which binds F-actin without stabilizing it in living cells and tissues[41]. Consistent with the cell shape and F-actin levels, the neural plate cells in control morphants showed a dynamic process that generates contractile actin bundling (Fig. 2c and Supplementary Movies 3 and 4). EphrinB2 or Ror2 morphants displayed a dramatic reduction in contractile actin bundle formation. The defects were rescued by re-introducing ephrinB2-WT or Ror2-WT, but not the ephrinB2 (ephrinB2-Δ4) or Ror2 (Ror2-ΔFr, or ΔPR) interaction mutants (Fig. 2c and Supplementary Movies 3 and 4).

Taken together, our results suggest that the interaction between ephrinB2 and Ror2 is required for contractile actin bundle formation.

## Wnt4 regulates apical constriction during neural tube closure

A study in the *Xenopus* embryo demonstrated that Wnt4 MO-mediated knockdown causes severe gastrulation defects and an open neural tube in a dose-dependent manner[39]. In *Drosophila*, Wnt4 is also known to provide an instructive regulatory cue for PCP axis establishment[10]. Thus, we sought to test the association of ephrinB2 with Wnt4, a candidate found in the ephrinB2 immune complex, and whether there is specificity among Wnt ligands for the association. Co-IP from embryos exogenously expressing ephrinB2 along with six different

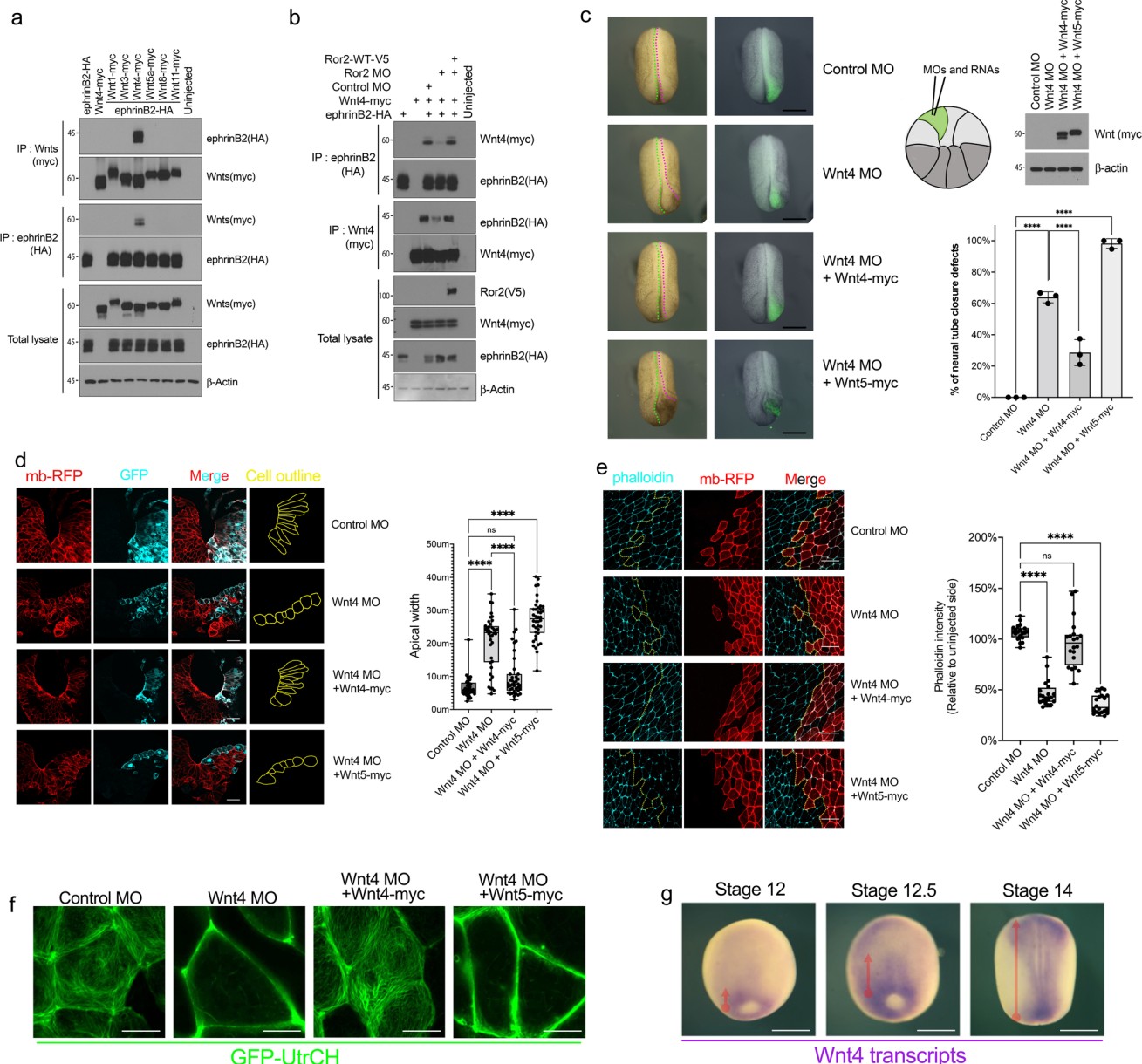

**Fig. 3 | Wnt4 interacts with ephrinB2 via Ror2 to regulate apical constriction.**
**a** Co-IP assay with ephrinB2 (500 pg) and Wnt ligands (100 pg). **b** Co-IP assay with
Wnt4 (100 pg) and ephrinB2 (100 pg) in Ror2 morphants. **c** Representative embryo
morphology by Wnt4 knockdown (2 ng). See also Supplementary Movie 1. Sche-
matic of D1.1 microinjection in the 16-cell stage embryo. Membrane-GFP indicates
injected side. The lateral boundaries of the neural plate are marked by dotted lines
(green; uninjected side and magenta; injected side). Immunoblotting represents
expression levels of Wnt4 and Wnt5a (50 pg). Data means SEM. Numbers of
embryos/experiments: 55/3, 64/3, 57/3, and 57/3 from left to right for each column.
One-way ANOVAs followed by Dunnett's multiple comparison tests were used. n.s.,
not significant, ****$p < 0.0001$. Scale bar, 500 μm. **d** Transverse section through the
neural plate. Unilateral injection of MO and RNA as indicated. Membrane-RFP

labeled outline of the neural plate cells. GFP (Cyan) indicates the injected side. The
box plot shows Min to Max with all data points. Numbers of embryos/experiments:
40/4 for each column. One-way ANOVAs followed by Dunnett's multiple compar-
ison tests were used. n.s., not significant, ****$p < 0.0001$. Scale bar, 50 μm. **e** Apical
actin levels in the neural plate by Phalloidin staining. Membrane-RFP indicates the
injected side. The box plot shows Min to Max with all data points. Numbers of
embryos/experiments: 20/4 for each column. One-way ANOVAs followed by Dun-
nett's multiple comparison tests were used. n.s., not significant, *$p < 0.005$, and
****$p < 0.0001$. Scale bar, 50 μm. **f** Stills from time-lapse imaging of GFP-UtrCH
(100 pg) overexpressed in neural plates from stage 14 to 15 embryos. See also
Supplementary Movie 6. Scale bar, 10 μm. **g** Spatial expression pattern of Wnt4.
WISH of stage 12, 12.5, and 14 embryos with anti-Wnt4 probes. Scale bar, 500 μm.

Wnt ligands indicated that ephrinB2 is specifically associated with
Wnt4 (Fig. 3a). To find the region within ephrinB2 necessary for the
interaction with Wnt4, Co-IP was performed with the previously
described ephrinB2 deletion mutants (Supplementary Fig. 1a). Unex-
pectedly, the secreted Wnt4 molecule also requires the ephrinB2
intracellular domain to associate with ephrinB2, rather than the
extracellular domain (Supplementary Fig. 2a). The extracellular
Frizzled-like domain (Fr) of Ror2 is known to interact with Wnt

ligands[42–45], and therefore begs the question of whether the interaction
between ephrinB2 and Wnt4 is mediated by Ror2. To test this
hypothesis, we examined the ephrinB2/Wnt4 interaction in Ror2
morphants. Loss of Ror2 significantly reduced the ephrinB2/Wnt4
interaction, which was rescued by re-introducing MO-resistant Ror2-
wild-type mRNA (Fig. 3b). Furthermore, co-expression of Ror2
increased the association between Wnt4 and ephrinB2 (Supplemen-
tary Fig. 2b), and reciprocally, co-expression of Wnt4 increased the

ephrinB2/Ror2 interaction (Supplementary Fig. 2c). Of note, another Ror2 ligand, Wnt5, did not enhance the interaction between ephrinB2 and Ror2 (Supplementary Fig. 2c), indicating specificity towards Wnt4. To gain some insight into what may regulate the specificity of the Wnt4 interaction with ephrinB2, we examined whether Ror2 specifically binds Wnt4. Our Co-IP results from embryos overexpressing Ror2 and several Wnts clearly showed that Ror2 interacts with both Wnt4 and Wnt5a (non-canonical Wnt ligands), not Wnt8 (canonical Wnt ligand) (Supplementary Fig. 2d). However, we did not observe a significantly different binding affinity for Ror2 between Wnt4 and 5 (Supplementary Fig. 2d). These results suggest a couple of possibilities regarding why ephrinB2 is preferentially found in a complex with Wnt4: 1) Wnt4 binding may modulate a Ror2 conformational change that results in interactions with specific binding partners, leading to formation of the WERDS complex; 2) Wnt4 induces phosphorylation of Ror2 on sites, which modulate its binding partner preferences. Regardless, our results indicate that Wnt4 associates with ephrinB2 via Ror2.

Having established that a Wnt4/Ror2/ephrinB2 complex may exist, we tested whether there is a functional relationship between these proteins; specifically, whether Wnt4 is also required for proper neural tube closure. To avoid broader morphogenetic defects that would obscure an effect on neural tube formation, the Wnt4 MO dosage was carefully titrated to facilitate studying the role of Wnt4 in the process of neural tube closure. We determined that injection of 1.5 ng of Wnt4 MO targeted to D1.1 caused NTDs in approximately 60% of embryos without causing gastrulation defects (Supplementary Fig. 2e). Thus, 1.5 ng of Wnt4 MO along with GFP mRNA (as a tracer) was injected into the D1.1 blastomere. Knockdown of Wnt4 clearly caused NTDs (Fig. 3c and Supplementary Movie 1), while re-expressing Wnt4 in the morphants rescued neural tube closure (Fig. 3c). In contrast, expressing an equivalent amount of Wnt5, a ligand for Ror2, does not rescue the Wnt4 MO-mediated defect (Fig. 3c), confirming the specific requirement for Wnt4. To further validate the loss-of-function phenotype conferred by the Wnt4 MO, we employed a knockout strategy using CRISPR/Cas9. Wnt4 CRISPR-targeted embryos also show neural tube closure defects confirming that Wnt4 regulates this process (Supplementary Fig. 2f and Supplementary Movie 5).

To determine whether loss of Wnt4 affects apical constriction, the shape of neural plate cells was analyzed. Examination of neural plate cross-sections showed that depletion of Wnt4 dramatically impaired apical constriction (Fig. 3d). Control MO containing neural plate cells displayed a normal neural groove and narrow apical width (<10 μm). Similar to ephrinB2 or Ror2 morphants, the neural plate cells harboring the Wnt4 MO showed a significantly expanded apical width (20–30 μm). This increase in apical width upon loss of Wnt4 was rescued by re-expression of Wnt4, but not Wnt5 (Fig. 3d). Furthermore, the phalloidin staining showed a significant decrease in apical actin levels upon loss of Wnt4 (Fig. 3e). This Wnt4 MO-mediated decrease in apical F-actin level was rescued by re-introduction of Wnt4, but not Wnt5 (Fig. 3e).

Another event associated with apical constriction is the phosphorylation of myosin light chain (p-MLC), which results in actomyosin cable contraction[46]. Consistent with the apical F-actin levels observed in morphants (Figs. 2b and 3e), knock down of endogenous ephrinB2, Ror2, or Wnt4 significantly decreased the number of p-MLC-positive cells in the neural plate of embryos (Supplementary Fig. 2g). Re-expression of wild-type ephrinB2, Ror2, or Wnt4 in the morphants rescued the number of p-MLC-positive cells in the neural plate. However, expression of the interaction mutants (ephrinB2-Δ4, Ror2-ΔFr, Ror2-ΔPR) or Wnt5 in lieu of Wnt4 failed to rescue the number of p-MLC-positive cells (Supplementary Fig. 2g). Live-cell imaging for apical actin bundling via exogenous expression GFP-UtrCH in the neural plate provided similar results. Loss of Wnt4 significantly decreased the generation of contractile actin filaments, however, this defect was rescued by re-introducing Wnt4, but not by Wnt5 (Fig. 3f and

Supplementary Movie 6). These results indicate that Wnt4 is involved in apical constriction process during neural tube closure.

During neural tube closure, the neural plate acquires progressive apical constriction from the posterior to anterior portions of the embryo. Butler and Wallingford used time-lapse confocal images of the late gastrula to neurula stages and showed that initially posterior neural plate cells contracted in the apical area and then moved more anteriorly[7]. To confirm this result, phospho-MLC levels in neural plate cells were analyzed using stage 13 to stage 17 Xenopus embryos. Consistent with these previous morphological observations[7], we observed p-MLC-positive cells only in the posterior of the embryo at stage 13 and then p-MLC-positive cells expanded toward the anterior region, suggesting that the neural plate acquires apical constriction progressively from posterior to anterior in the embryo (Supplementary Fig. 2h). Interestingly, Wnt4 temporal and spatial expression showed a similar pattern. At stage 12, Wnt4 transcripts were detected only in the posterior of the neural plate, and then Wnt4 expression expanded anteriorly during the neural stage (Fig. 3g). This result is consistent with Wnt4 being involved in progressive apical constriction.

## Dsh2 mediates the formation of a Wnt4, EphrinB2, Ror2, Dsh2, and Shroom3 (WERDS) complex

Although the Co-IP analysis established an association between ephrinB2 and its binding partners Ror2 and Wnt4, the question remained whether ephrinB2 and the binding partners regulate apical constriction as a signaling complex or whether each interaction is involved in apical constriction independently. One known ephrinB2-binding partner, Disheveled (Dsh), is a key scaffold component in Wnt signaling[47], and raises the possibility that ephrinB2 and its binding partners form a signaling complex. There are several lines of evidence to support this possibility: 1) various studies have shown that Dsh interacts with ephrinBs[48] as well as other ephrinB2 immune-complex members, including Ror2[49], and Shroom3[20]; 2) the last four to six amino acids of ephrinB2, necessary for the interaction with Ror2, are also required for the ephrinB2/Dsh2 interaction (Fig. 1f); 3) the Proline-rich domain of Ror2, necessary for the interaction with ephrinB2, is also required for the binding of Ror2 with Dsh2 (Fig. 1g)[44]; 4) in vertebrate systems, loss-of-function and mutation analyses of Dsh have verified a role in NTDs[36]. Thus, we asked whether Dsh2 mediates the interaction between ephrinB2 and Ror2. To address this question, we examined the ephrinB2/Ror2 interaction in Dsh morphants. Interestingly, loss of endogenous Dsh abolished the interaction between exogenously expressed ephrinB2 and Ror2, and re-introduction of WT-Dsh2 rescued the interaction (Fig. 4a). This result clearly indicates that Dsh2 mediates the ephrinB2/Ror2 interaction.

Since previous studies suggest that ephrinB2 and Ror2 bind to the C-terminal region of Dsh2, which also includes the DEP domain[48,49], we generated serial deletion mutants of Dsh2 to determine the specific region within Dsh2 necessary for the interaction with ephrinB2 and Ror2. Co-IP analysis with serial deletion mutants of Dsh2 showed that the last eight amino acids (PDZ-binding motif; PBM) are necessary for the interaction with ephrinB2, and a region of 43 amino acids upstream from the PBM that we termed the Tail 9 (T9) region is necessary for the Ror2 interaction (Fig. 4b–d and Supplementary Fig. 3a–c), however, the T9 region is not required for the interaction between ephrinB2 and Dsh2 (Supplementary Fig. 3d). Our Co-IP analyses suggest that ephrinB2 and Ror2 bind to distinct regions of the Dsh2 C-terminus (PBM and T9, respectively). Since our ephrinB2 and Ror2 constructs have an HA or V5 tag at the C-terminus, we verified that the C-terminal tag does not affect the ephrinB2/Dsh2 interaction. We constructed N-terminal tagged ephrinB2 by inserting an HA tag after the signal peptide (SP). Co-IP analysis from embryos expressing GFP-Dsh2 and N- or C-terminal tagged ephrinB2 reveals that the C-terminal tag does not hinder the interaction between Dsh2 and ephrinB2 (Supplementary Fig. 3e). We also tested whether the C-terminal tag disturbs the

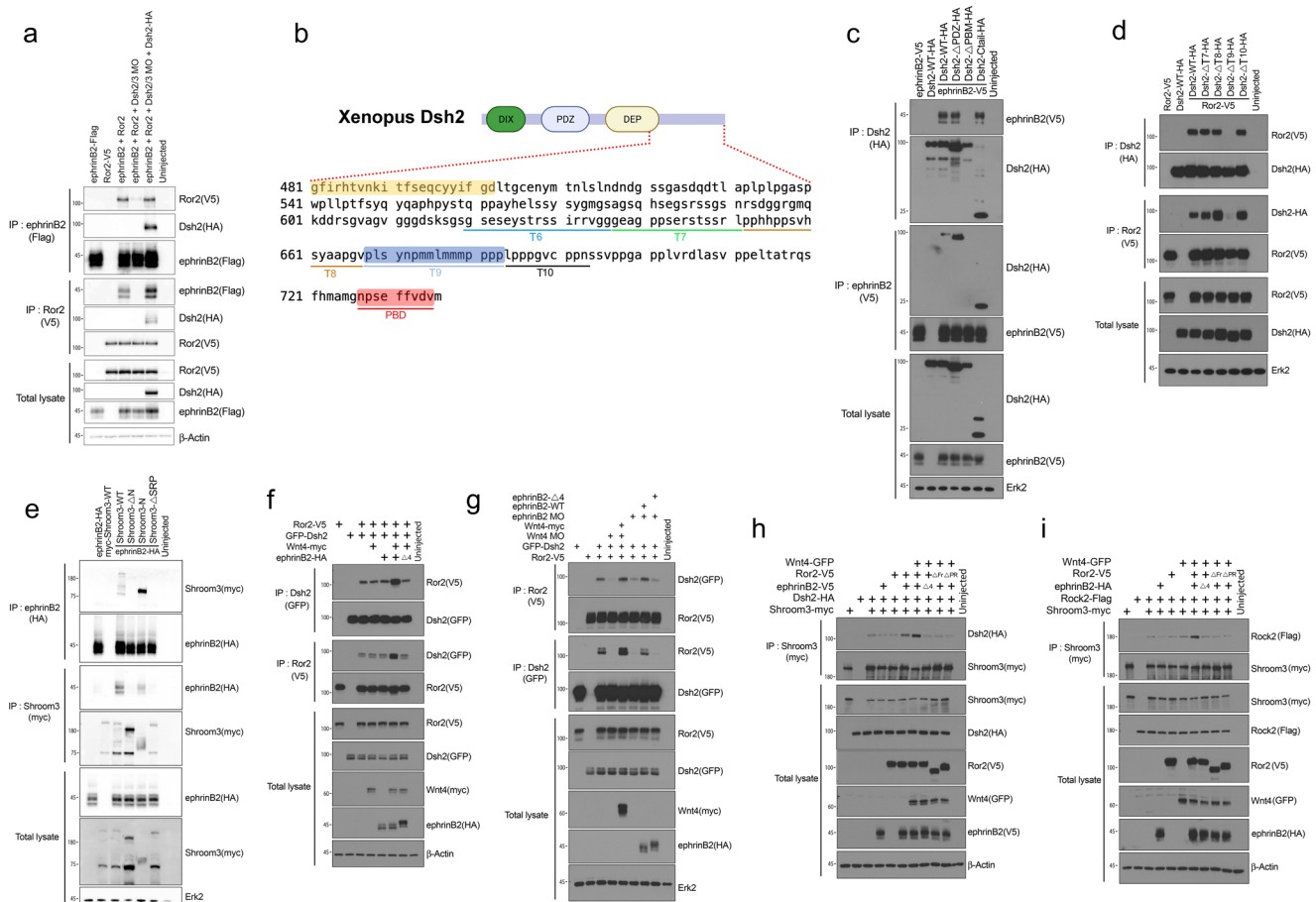

**Fig. 4 | Wnt4, ephrinB2, Ror2, Dsh2, and Shroom3 form a signaling complex.**
**a** Co-IP assay with ephrinB2 (100 pg) and Ror2 (100 pg) in Dsh2/3 morphants.
**b** Schematic of Dsh2 serial deletion in the tail region (Ctail). **c** Co-IP assay with
ephrinB2 (500 pg) and Dsh2 (1 ng) deletion mutants. **d** Co-IP assay with Ror2 (1 ng)
and Dsh2 (1 ng) deletion mutants. **e** Co-IP assay with ephrinB2 (500 pg) and con-
served domain deletion mutants of Shroom3 (1 ng). **f** Co-IP assay with Ror2 (100 pg)
and Dsh2 (200 pg) in the presence of exogenously expressed ephrinB2 (25 pg) and
Wnt4 (50 pg). **g** Co-IP assay with Ror2 (100 pg) and Dsh2 (200 pg) in neural plates of
ephrinB2 or Wnt4 morphants. **h** Co-IP assay with Dsh2 (200 pg) and Shroom3
(100 pg) in the presence of other WERDS components, Wnt4 (50 pg), Ror2 (100 pg),
and ephrinB2 (25 pg). **i** Co-IP assay with Shroom3 and Rock in presence of other
WERDS components, Wnt4 (50 pg), Ror2 (100 pg), and ephrinB2 (25 pg).

intramolecular interaction of the Dsh2 C-terminus with its PDZ
domain. Our Co-IP from embryos expressing GFP-Dsh2-ΔPBM (lacking
the PDZ-binding motif) and N- or C-terminal tagged Dsh2 tail con-
structs (consisting of just the C-terminal 234 amino acids) demon-
strated that the small C-terminal tag (HA, 8 amino acids) does not
hinder the interaction between the PDZ domain and C-terminal PBM in
Dsh2 (Supplementary Fig. 3f).

Previous studies in mice have shown that disruption of Shroom3
via gene trap mutagenesis causes neural tube closure defects[23] and
pathogenic mutations in humans are associated with anencephaly[40,50].
Furthermore, a previous report has proposed that Dsh2 regulates
Shroom3 functions through an interaction with the DEP domain within
Dsh2[20,24]. Our Co-IP also confirmed that the DEP domain in Dsh2 is
necessary for the Dsh2/Shroom3 interaction (Supplementary Fig. 3g).
Human and mouse Shroom3 encodes two isoforms, *shrmL* containing
a PDZ domain and *shrmS* without a PDZ domain[23,51]. However, *Xenopus*
and *Zebrafish* Shroom3 encode only the *shrmS* isoform (Supplemen-
tary Fig. 3h). Despite the ascribed role of the PDZ domain in associa-
ting with binding partners, comparison analysis demonstrated that the
*shrmL* and *shrmS* isoforms showed similar functional activities[46] and
the molecular function of the PDZ domain of Shroom3 has remained
elusive. Although *Xenopus* Shroom3 lacks a PDZ domain, the serine-
proline-rich region (SPR) is highly conserved as well as the ASD1 and
ASD2 domains (Supplementary Fig. 3h)[20]. However, the molecular
function of SPR is still poorly understood. In order to verify the

possible interaction between Shroom3 and the protein complex con-
taining ephrinB2, we constructed domain deletion mutants of
Shroom3 (Supplementary Fig. 3i) and performed co-expression and
Co-IP analysis in embryos. Interestingly, truncation of the SPR within
Shroom3 significantly decreased the interaction between ephrinB2
and Shroom3 (Fig. 4e). Taken together, our results suggest that Dsh2 is
a central scaffold in the formation of a complex that also includes
Wnt4, ephrinB2, Ror2, and Shroom3.

Recent studies suggest that the binding pocket of the PDZ-
domain of Dsh is occupied by its intrinsic highly conserved C-terminus
(NPCEFFVDVM)[52,53]. In the uninduced state, the C-terminus binds to the
PDZ domain to form a "closed conformation"[52,53]. The conformational
change from a closed to open state regulates Dsh functions in Wnt/β-
catenin and Wnt/planar cell polarity (PCP) signaling pathways[52]. Since
our results showed that ephrinB2 binds to the C-terminus of Dsh2, we
hypothesized that the interaction of ephrinB2 and Dsh2 induces the
"open conformation" and exposes the Ror2 binding region of Dsh2,
resulting in the enhancement of the Ror2/Dsh2 interaction. As a test of
this concept, the Ror2/Dsh2 interaction was evaluated in embryos also
overexpressing either ephrinB2 or Wnt4 or both. Co-IP analysis
showed that the interaction of Ror2 and Dsh2 is enhanced by co-
expression of ephrinB2 and Wnt4 (Fig. 4f). However, a mutant
ephrinB2 (ephrinB2-Δ4) that lacks the ability to interact with Dsh2 does
not increase the Ror2/Dsh2 interaction. In neural plate explants, MO-
mediated knockdown of Wnt4 or ephrinB2 dramatically reduced the

interaction of Ror2 and Dsh2, and the interaction was recovered by re-expression of wild-type Wnt4 or ephrinB2, respectively (Fig. 4g). In addition, the overexpression of Wnt4, ephrinB2, and Ror2 significantly enhanced the interaction of Dsh2 with Shroom3 (Fig. 4h) and Shroom3 with Rock (Fig. 4i). As expected, the ephrinB2 and Ror2 interaction mutants (ephrinB2-Δ4, Ror2-ΔFrz, or Ror2-ΔPR) do not increase these associations.

To further test our hypothesis, we employed a Dsh2-S267E mutant. A previous study demonstrated that Dsh3 also has an interaction between the C-terminal PBM (PDZ-binding motif) and its own PDZ domain[54]. This interaction can be regulated by post-translational modifications including phosphorylation. Interestingly, the point mutation (Serine 263 to Glutamic acid) suppressed the intramolecular interaction of the PDZ domain and the C-terminal motif resulting in the open conformation of Dsh3[54]. Since Dsh2 and 3 have a highly conserved PDZ domain and PBM, we made the same single amino acid mutation (S267E) in Dsh2 to test whether the open conformation of Dsh2 is more efficient at interacting with WERDS components. Co-IP analysis showed that the Dsh2-S267E mutant has a more robust interaction with ephrinB2 and Ror2 than wild-type Dsh2 (Supplementary Fig. 3j and k). These results support our model that ephrinB2 binds Dsh2 resulting in the open conformation of Dsh2, which enhances the interaction with Ror2.

Taken together, our results suggest that ephrinB2 induces a conformational change in Dsh2 that enhances the formation of a Wnt4-ephrinB2-Ror2-Dsh2-Shroom3 (WERDS) protein complex.

## EphrinB2 interaction with Dsh2 can suppress canonical Wnt signaling

Wnt signaling can be divided into canonical Wnt/β-catenin signaling and non-canonical Wnt/PCP signaling[47]. Numerous studies have demonstrated that canonical and non-canonical Wnt signaling can share ligands, Frizzled receptors, and Dsh scaffolding proteins[55]. Interestingly, several studies have shown that non-canonical Wnt signaling antagonizes the canonical Wnt signaling in developing *Xenopus* embryos and cancer cells[56,57]. However, the mechanism of this mutually exclusive relationship between canonical and non-canonical Wnt signaling has been elusive. In our study, we hypothesized that the ephrinB2 bound C-terminus of Dsh2 results in a conformational change of Dsh2 from the "closed conformation" to an "open conformation". Interestingly, a couple of studies have suggested that a conformational change of Dsh regulates its functions by modulating accessibility of binding partners[52,53]. To further test our hypothesis, we examined functional features associated with activation of the canonical Wnt/β catenin signaling pathway. We examined the effect of ephrinB2 on canonical Wnt-associated events such as nuclear localization of β-catenin (Supplementary Fig. 4a). As expected, neither non-canonical Wnt5 ligand alone nor ephrinB2 alone induced the nuclear localization of β-catenin, while canonical Wnt ligand, Wnt3, induced nuclear localization of β-catenin. Interestingly, co-expression of ephrinB2-WT dramatically suppressed the β-catenin nuclear localization induced by Wnt3 but ephrinB2-Δ4, which cannot bind to Dsh2, did not suppress this localization (Supplementary Fig. 4a). These results suggest that ephrinB2 represses canonical Wnt signaling by interacting with Dsh2 and supports the concept that ephrinB2/Dsh2 acts via non-canonical Wnt signaling.

To further test the concept that an ephrinB2 interaction with Dsh2 may favor non-canonical Wnt/PCP signaling rather than canonical Wnt signaling, we designed experiments based upon the ephrinB2 interaction with Dsh2 and took advantage of the developing *Xenopus* embryo. Various studies have shown that ephrinB2 is tyrosine phosphorylated in its C-terminus upon binding the cognate Eph receptor or in *cis* through activation of the FGF receptor, which inhibits the interaction of Dsh2 and ephrinB2[34,48,58]. Since ephrinB2 functions may be suppressed by endogenous Eph receptors, which are expressed in

the VMZ region at the early blastula stage[59], we employed ephrinB2 C-terminal tyrosine mutants to preclude these Eph receptors from phosphorylating ectopically expressed ephrinB2. Previously, we demonstrated that phosphorylation of tyrosines 324 and 325 in ephrinB1 disrupts the ephrinB1/Dsh interaction[58]. Since ephrinB1 and ephrinB2 have a highly conserved intracellular domain, we tested whether phosphorylation of tyrosines in the PDZ-binding motif (PBM) of ephrinB2 also affects the ephrinB2/Dsh interaction. Consistent with the ephrinB1/Dsh2 interaction, Co-IP analyses confirmed that ephrinB2-Y2E mutant (phosphomimetic mutant) did not interact with Dsh2 but the Y2F mutant (non-phosphorylatable mutant) strongly interacts with Dsh2 (Supplementary Fig. 4b).

Wnt/β-catenin signaling plays a critical role in dorsoventral axis patterning and many studies have shown that ectopic expression of a canonical Wnt ligand in the ventral side of an embryo induces a secondary axis[60,61]. Therefore, in our experiments, we overexpressed Wnt3 at the ventral marginal zone (VMZ) to induce axis duplication. We exogenously expressed ephrinB2 wildtype and tyrosine mutants (ephrinB2-Y2F or -Y2E) along with Wnt3 in the VMZ (Supplementary Fig. 4c). EphrinB2-WT slightly reduced secondary axis formation (approximately 20% reduction) while expression of the neural plate marker *Sox2* was markedly decreased. Notably, the ephrinB2-Y2F mutant that effectively maintains an interaction with Dsh2, dramatically reduced secondary axis formation and *Sox2* expression (Supplementary Fig. 4c). In contrast, ephrinB2-Y2E, a phosphomimic mutant that does not interact with Dsh2, failed to reduce axis duplication and *Sox2* expression induced by ectopic Wnt3 expression (Supplementary Fig. 4c). Further support for this result came from examining a direct target gene of Wnt/β-catenin signaling, *Siamois*, which was also examined in these ectodermal explants (Supplementary Fig. 4d). Consistent with the axis duplication results, co-expression of ephrinB2-Y2F dramatically decreased *Siamois* expression induced by Wnt3. As expected, ephrinB2-Y2E did not affect *Siamois* expression levels in the embryos.

Ror2 has been reported to be a receptor tyrosine kinase (RTK)[44], although recent studies indicate it is a pseudo-kinase[62,63]. Since ephrinB2 has been shown to modulate certain downstream signaling events through phosphorylation at tyrosine residues within the intracellular domain[35], we tested whether Wnt ligands induce the tyrosine phosphorylation of ephrinB2. Six different tagged Wnt ligands representing canonical and non-canonical Wnts were overexpressed along with ephrinB2 in embryos. Western blot analysis was performed on ephrinB2 immune complexes using specific antibodies for ephrinB2 tyrosine phosphorylation. Constitutively active FGF receptor, a positive control, induced phosphorylation at three tyrosine sites. However, none of the Wnt ligands induced tyrosine phosphorylation of ephrinB2 (Supplementary Fig. 4e). In addition, we also confirmed that overexpressing Ror2 along the Wnt4 ligand did not induce tyrosine phosphorylation of ephrinB2 (Supplementary Fig. 4f).

Taken together, our results suggest that ephrinB2 antagonizes canonical Wnt signaling and supports the concept that ephrinB2 regulates a conformational change of Dsh2 by interaction with the PBM that functions via the non-canonical Wnt signaling pathway.

## EphrinB2 and Ror2 localize at tricellular junctions and regulate Shroom3 function

Shroom3 is known to localize to apical junctions and recruit its binding partners to modulate actomyosin contraction. Interestingly, previous studies have reported that overexpressed Shroom3 is enriched at apical junctions[22,64], and the N-terminal region of Shroom3 that includes a PDZ domain is required for this specific subcellular localization[64]. Additionally, a known Shroom3 binding partner, Wtip, also enriched at the tricellular junctions, which is specialized structure has been identified at the place where three cells meet in vertebrate epithelial cells[65,66]. Therefore, it was of interest to assess in our system

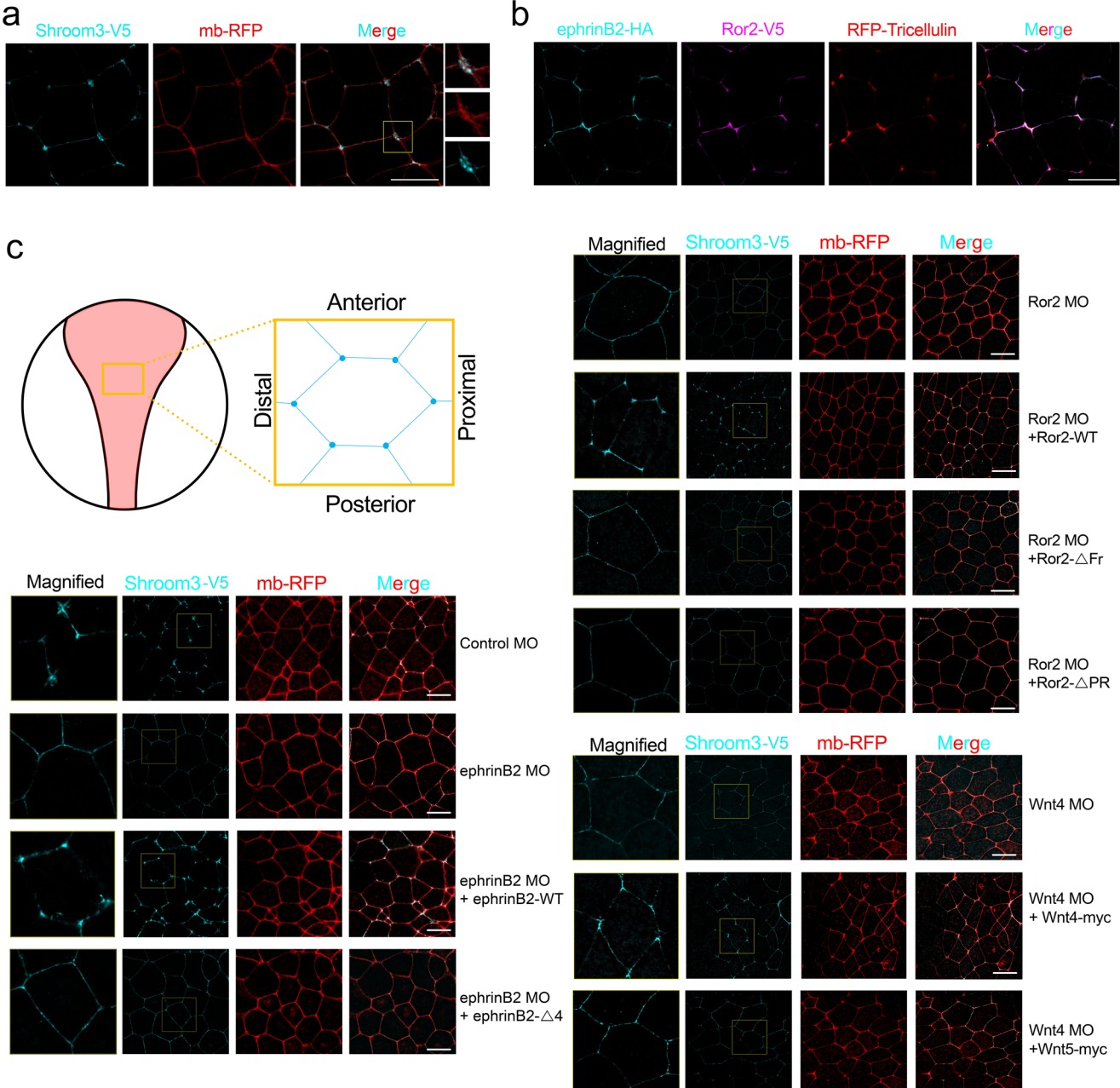

**Fig. 5 | EphrinB2 and Ror2 regulate Shroom3 accumulation at tricellular junctions. a** Representative images of V5-Shroom3 (100 pg) subcellular localization in neural plate cells in stage 15 embryo. Membrane-RFP (100 pg) labeled cell boundaries. The yellow box indicates the magnified area. Scale bar, 10 μm. **b** Representative images of ephrinB2 (25 pg) and Ror2 (50 pg) subcellular localization in neural plate cells in stage 15 embryo. RFP-Tricellulin (50 pg) labeled tricellular junctions. Scale bar, 10 μm. **c** Representative images of Shroom3 localization in neural plate cells of ephrinB2, Ror2, or Wnt4 morphants. Membrane-RFP (100 pg) labeled cell boundaries. Yellow boxes indicate the magnified area. Scale bar, 20 μm.

(*Xenopus* neural plate explants) whether Shroom3 along with other complex members (ephrinB2 and Ror2) similarly localize to apical junctions. Since antibodies against endogenous *Xenopus* Shroom3 are not available, we conducted a titration experiment to determine the lowest possible dose of mRNA that allowed imaging in the *Xenopus* neural plate. Under these conditions, we found that Shroom3 localized in the proximity of tricellular junctions using GFP-Tricellulin as a tricellular junction marker[65] (Supplementary Fig. 5a). This localization was more evident in higher resolution images (Fig. 5a). Coimmunostaining of exogenously expressed tagged versions ephrinB2 and Ror2 along with GFP-Tricellulin confirm that ephrinB2 and Ror2 also localized at the tricellular junctions (Fig. 5b). Moreover, this localization was confirmed by immunostaining of endogenous

ephrinB2 and Ror2 that also showed these two proteins are enriched at tricellular junctions (Supplementary Fig. 5b).

Our results showing a greater enrichment of Shroom3, ephrinB2, and Ror2 at the tricellular junctions prompts the question of whether this localization of Shroom3 is induced during apical constriction and whether it is dependent upon the other complex members. Thus, we used MO-mediated knockdown of endogenous ephrinB2, Ror2, or Wnt4 and observed that exogenous Shroom3 localization at tricellular junctions is suppressed in neural plate explants (Fig. 5c). This suppression was relieved by re-expressing the wild-type counterparts of these target proteins (Fig. 5c). However, the ephrinB2 mutant that does not interact with Dsh2 (ephrinB2-Δ4), and the Ror2 mutants that either cannot bind Wnts or Dsh2 (Ror2-ΔFrz, Ror2-ΔPR, respectively) failed to

rescue junctional enrichment of Shroom3 (Fig. 5c). Moreover, expressing Wnt5 instead of Wnt4 in a Wnt4 morphant explant did not rescue localization of Shroom3 to tricellular junctions (Fig. 5c). These results suggest that the appropriate interactions among the Wnt4/ ephrinB2/Ror2/Dsh2 (WERD) signaling complex members regulate Shroom3 subcellular localization during apical constriction.

## WERDS signaling complex regulates apical constriction

Although the loss of Shroom3 function in animal models, as well as humans, is associated with NTDs[26,40], we also determined whether the same is true in our system with MO-mediated loss of Shroom3 (Supplementary Fig. 6a). In the previous experiments using biochemistry, MO-mediated knockdown and replacement experiments with wild-type and interaction mutants, we have shown the requirement for the function and interaction among WERDS complex members for neural tube closure. As a further test of the WERDS complex model, we implemented a complementary approach that uses an exogenous expression assay of apical constriction. Several studies have shown ectopic expression of Shroom3 is sufficient to induce ectopic apical constriction resulting in pigment accumulation in the *Xenopus* ectoderm[21,66]. First, we determined that exogenous expression of more than 200 pg of Shroom3 mRNA induced ectopic apical constriction in the ectodermal layer as evidenced by pigment accumulation (Supplementary Fig. 6b), while expression of only 100 pg of Shroom3 mRNA did not (Supplementary Fig. 6b). To test our model, 100 pg of Shroom3 mRNA was injected along with various combinations of other WERDS complex components except Dsh2, where we relied upon the abundant endogenous Dsh2 expression[67]. In this carefully titrated assay, only when all other components were exogenously expressed did the embryos display apical constriction (Fig. 6a–d and Supplementary Fig. 6c–f). As a control, high doses of each single component along with Shroom3 did not induce apical constriction (Fig. 6a and Supplementary Fig. 6c). As expected from our model, unlike wild-type versions of ephrinB2 and Ror2, the interaction mutants (ephrinB2-Δ4, Ror2-ΔFr, or Ror2-ΔPR) did not induce apical constriction (Fig. 6b–d and Supplementary Fig. 6d–f). These results indicate that Wnt4, ephrinB2, Ror2, Dsh2, and Shroom3 regulate apical constriction as a signaling protein complex.

During apical constriction, Shroom3 interacts with the Rho kinase, ROCK, which phosphorylates and activates Myosin light chain (phospho-MLC) and induces apical actin filament formation[22]. As a further examination of the WERDS complex, we performed immunostaining for phospho-MLC and phalloidin staining in the ectodermal explants expressing the various combinations of WERDS complex members. Our results showed that only the simultaneous exogenous expression of all the WERDS components significantly elevated the level of phospho-MLC and actin filaments, along with pigment accumulation at the ectopic apical constriction sites in explants (Fig. 6e). Collectively, our data suggest that ephrinB2 plays a substantive role along with the other WERDS members Wnt4, Ror2, Dsh2, Shroom3 to regulate apical constriction as a multiprotein signaling complex (Fig. 6f).

## Discussion

Apical constriction is a morphogenetic process that contributes to a variety of developmental events including neural tube formation. The contraction of actin-myosin networks and cell-cell adhesion drives apical constriction resulting in cell shape changes as well as tissue morphology transformations. Shroom3 is a key molecule for this actomyosin contractility in the developing vertebrate embryo and localizes to apical junctions where it recruits Rock, resulting in the formation of circumferential actin-myosin bundles and increasing contractile tension[20–22,26,40]. Although various studies suggest that initiating the formation of actin-myosin networks is tightly regulated by developmental time and cell positioning, the mechanisms of how

the cells initiate apical constriction remain unknown. Here, we uncovered a signaling protein complex, named WERDS, that regulates contractile actin-myosin network formation during neural tube closure. However, there are still open questions that need to be addressed in future studies.

In this work, using Co-IP combined with mass spectrometry from *Xenopus* embryonic extracts overexpressing ephrinB2, we identified a signaling complex involved in apical constriction, the WERDS complex. However, the most common problem engaged in the Co-IP analysis is the loss of many interacting partners during the experiment, such as transmembrane proteins and weak affinity binding proteins. Therefore, it is possible that other unidentified binding partners interact with the WERDS protein complex and play a role in apical constriction. One of the possible unidentified members is Vangl2, where the loss of function causes neural tube closure defects in various vertebrate embryos[3,4,7,68–72]. Furthermore, a subset of WERDS complex members are known to be Vangl2 binding partners, specifically Ror2 and Dsh2[37,55,73,74]. In addition to Vangl2, other proteins have been shown to play a role in apical constriction. For example, Lrp2, which has a functional interaction with Shroom3 and Gipc1 through its intracellular domain in the developing forebrain[75]. This report supports the idea that motifs within the intracellular domain of Lrp2 coordinate endocytic membrane removal for proficient apical constriction, and proper trafficking of PCP components to the appropriate positions during neurulation[75]. Another PCP component, Celsr1, is abundantly localized in adherens junctions with polarity towards the mediolateral axes of the neural plate. At these junctions, Celsr1 recruits Dsh (a WERDS complex member) and Frizzled receptors, which activate the PDZ-RhoGEF to upregulate Rock, triggering junctional actomyosin-dependent contraction in a planar-polarized manner[6]. It remains to be determined which of these proteins that have been shown to interact with various individual WERDS components may also be part of the larger WERDS signaling complex. A similar question arises from the small number of C-terminal amino acids required in ephrinB2 and Dsh2 for an interaction. It is possible that there is a direct interaction between the C-termini of these proteins and these last amino acids are required for the interaction. It is also possible that these C-terminal motifs are necessary to allow proper association among other portions of the proteins. Finally, perhaps an unidentified bridging protein between ephrinB2 and Dsh2 associates with these C-terminal motifs.

Further studies and structural analyses will be necessary to answer these questions.

Planar cell polarity signaling coordinates various morphogenetic cell movements and cell shape changes during neural tube closure. The polarized intracellular distribution of Wnt-PCP components, Vangl2, prickle, Celsr1, and Shroom3, regulates the cytoskeletal rearrangements and shrinking or elongation of cell-cell junctions resulting in mediolateral cell intercalation and apical constriction. Previous reports suggest anterior-posterior polarity signaling (PCP signaling) and apical-basal polarity signaling are coordinated and drive neural tube closure[6,22,71,72]. Our results showed that the F-actin levels (phalloidin signal intensity) enriched near all tricellular junction regions but were not antero-posteriorly polarized to one side of the cell (Figs. 2b and 3e). Generation of the contractile actin bundle formation (evidenced by live-cell imaging with GFP-UtrCH) appeared not to be polarized (Supplementary Movie. 3 and 4), nor did the subcellular distribution of ephrinB2 and Ror2. Furthermore, Shroom3 enrichment also did not display an anteroposterior polarized localization during apical constriction (Fig. 5c). Moreover, there are previous studies demonstrating that Vangl2 shows differing subcellular localization in phases of apical constriction versus mediolateral cell intercalation[69,70]. These results also raise the possibility that apical constriction and mediolateral cell intercalation processes are separate but normally coordinated events. Interestingly, recent work also proposes that apical constriction and

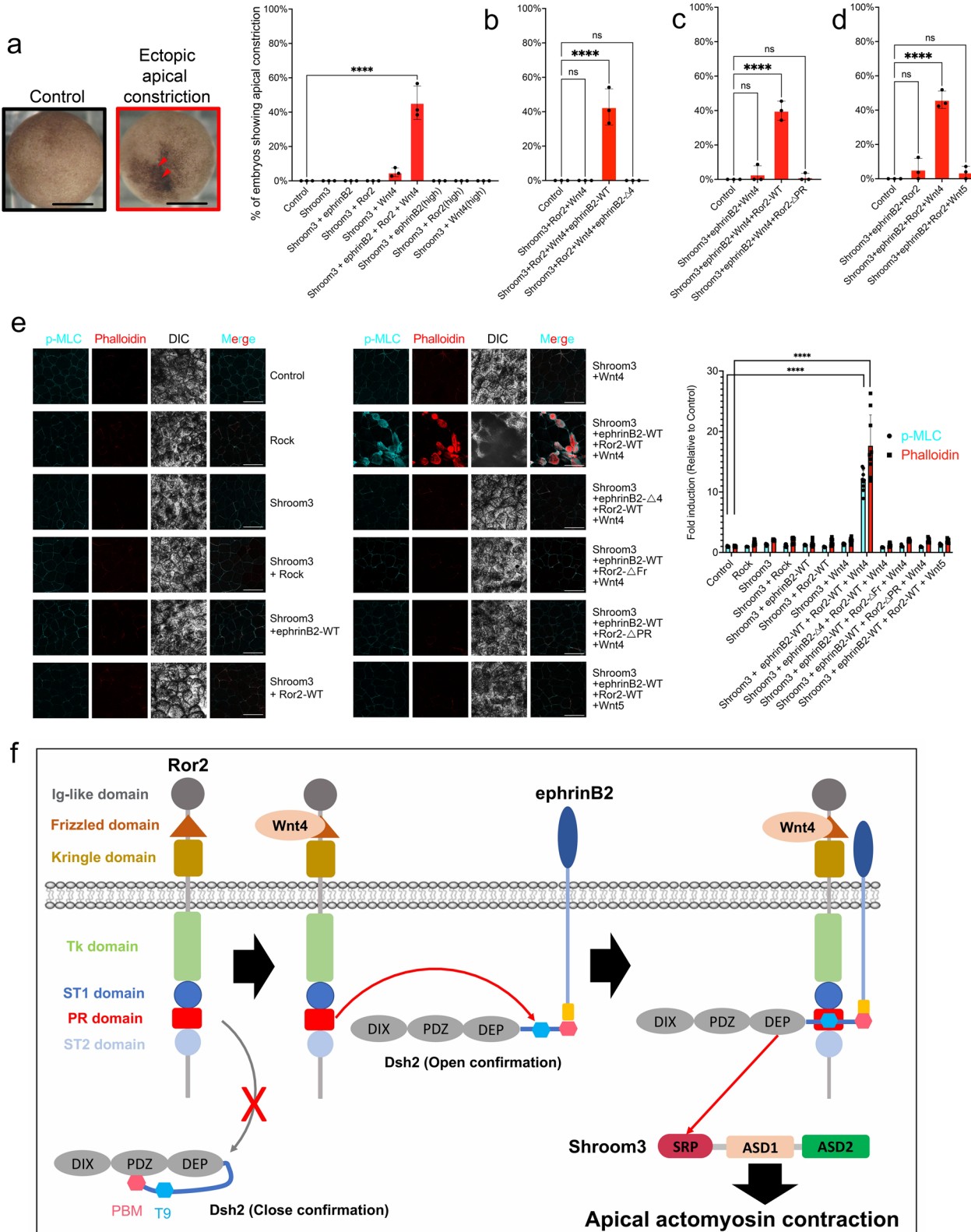

convergent and extension cell movements in the anterior neural plate are temporally and spatially separate processes[75].

Previous studies have shown that apical constriction is detectable in the lateral side of the neural plate at the beginning of neurulation, and it also seems to start from the posterior region and progress toward the anterior region of neural plate[7,68,69,76]. These results raise the question of whether there is a signal gradient to initiate apical

constriction. Interestingly, the progressive apical constriction is consistent with the Wnt4 expression pattern. Our HCR RNA-FISH showed that Wnt4 was expressed in the lateral side of the neural plate (Fig. 1b). Wnt4 transcripts were also detectable in the posterior neural plate in the early neurula embryo and Wnt4 expression was expanded to the anterior neural plate (Fig. 3g)[77,78]. Moreover, p-MLC-positive cells also displayed progressive expansion from posterior to the anterior neural

**Fig. 6 | WERDS signaling complex controls apical constriction. a** Ectopic apical constriction analysis with WERDS components. Representative images of ectopic apical constriction induced by exogenously expressed Shroom3 (200 pg). The red arrowheads indicate ectopic apical constriction in the ectoderm. Data means SEM. Numbers of embryos/experiments: 82/3, 89/3, 80/3, 77/3, 85/3, 102/3, 97/3, 91/3, 103/3, and 82/3 from left to right for each column. One-way ANOVAs followed by Dunnett's multiple comparison tests were used. n.s., not significant, ****$p < 0.0001$. Scale bar, 500 μm. See also Supplementary Fig. 6c. **b** Ectopic apical constriction analysis with the WERDS complex and ephrinB2-Δ4. Data means SEM. Numbers of embryos/experiments: 66/3, 68/3, 69/3, and 75/3 from left to right for each column. One-way ANOVAs followed by Dunnett's multiple comparison tests were used. n.s., not significant, ****$p < 0.0001$. See also Supplementary Fig. 6d. **c** Ectopic apical constriction analysis with the WERDS complex and Ror2-ΔPR. Data means SEM.

Numbers of embryos/experiments: 73/3, 70/3, 79/3, and 79/3 from left to right for each column. One-way ANOVAs followed by Dunnett's multiple comparison tests were used. n.s., not significant, ****$p < 0.0001$. See also Supplementary Fig. 6f. **d** Ectopic apical constriction analysis with the WERDS complex and Wnt5a. Data means SEM. Numbers of embryos/experiments: 75/3, 77/3, 78/3, and 78/3 from left to right for each column. One-way ANOVAs followed by Dunnett's multiple comparison tests were used. n.s., not significant, ****$p < 0.0001$. See also Supplementary Fig. 6g. **e** F-actin (Phalloidin) and phospho-MLC levels in ectodermal explants. Data means SEM. The number of ectodermal explants/experiments: 9/3 for each column. One-way ANOVAs followed by Dunnett's multiple comparison tests were used. n.s., not significant, ****$p < 0.0001$. Scale bar, 50 μm. **f** A model for WERDS signal complex.

plate during neural tube closure (Supplementary Fig. 2g). A recent study suggested that PCP is acquired progressively by a signal from the posterior region of the embryo[68]. This result raises the possibility that PCP establishment is involved in Wnt4 expression and progressive apical constriction. Further studies are required to understand how the apical constriction of neural plate cells is differentially activated.

Several studies have shown that overexpression of non-canonical Wnt ligands such as Wnt5a or Wnt11 antagonizes canonical Wnt signaling[56,57,79–81]. Wnt5 activation of the Wnt/Ca2+ pathway has been proposed to be a key factor for the antagonism toward Wnt/β-catenin signaling[80,81]. However, Topol et al. demonstrated that Wnt5 antagonizes Wnt/β-catenin signaling via *Siah2*, not Wnt/Ca2+ signaling. Although the evidence from gain and loss-of-function studies in the various model systems supports a function for Wnt5 in antagonizing Wnt/β-catenin signaling, the mechanism is unclear. In this study, our results clearly showed that overexpression of ephrinB2 suppresses Wnt3-induced β-catenin signaling (evidenced by suppression of nuclear β-catenin localization, secondary axis formation, and *Siamois* expression) via an interaction with Dsh2 (Supplementary Fig. 4a, c, and d). Our results raise the possibility that the non-canonical Wnt ligand induces an open conformation of Dsh2 that releases the C-terminal PDZ-binding motif from the PDZ domain of Dsh. This open conformation of Dsh may result in β-catenin degradation. Thus, further studies will be needed to clarify the mechanism of how non-canonical Wnt signaling antagonizes canonical Wnt signaling.

In our study, ephrinB2 drives actomyosin contractility through its interaction with other WERDS complex proteins. This is quite different from other morphogenetic processes where the EphB/ephrin-B system has been shown to regulate actomyosin contractility, such as during cell sorting. In this process, Eph/ephrin-mediated cell sorting has been shown to regulate tissue segregation and segmentation in various model systems[82,83]. The general mechanism is that tissue segregation occurs due to heterotypic contacts between unlike cells. These types of contacts display higher tension and less adhesion than homotypic contacts of like cells, and thus promote sorting[59,84,85]. Moreover, EphB/ephrin-B signaling activates actomyosin contractility via recruitment of myosin to the points of heterotypic contact, enhancing cortical tension[86]. In our study, neural tube closure is an ephrinB2-driven event through the interaction with Ror2, Wnt4 and Dsh2. A role for the binding of a cognate Eph receptor or an alternative receptor tyrosine kinase (RTK) functioning in *cis*, is currently unknown. However, it is clear from several studies[34,48,87] that Eph-mediated or alternative RTK-mediated tyrosine phosphorylation of ephrinB2 disengages Dsh2, which is the main scaffold of the WERDS complex. It will be of interest in future studies to see how such regulation may occur in vivo[88].

In summary, we identified a protein signaling complex, consisting of Wnt4, ephrinB2, Ror2, Dsh2, and Shroom3 (WERDS) that regulates actin-myosin contractility during neural tube closure in a vertebrate system. Further, the ephrinB2-Dsh2 interaction induces a conformational change of Dsh2 and facilitates the formation of the WERDS complex. Our results also show that the ephrinB2-Dsh2 interaction is enhanced by associating with the non-canonical Wnt receptor, Ror2, that is bound to the Wnt4 ligand. Gain and loss of function studies demonstrate that the WERDS signaling complex is localized at tricellular junctions and regulates the formation of actin-myosin networks that are critical for proper apical constriction during neural tube closure.

## Methods

### *Xenopus laevis*
Wild-type *Xenopus laevis* were obtained from Nasco, USA. All animal procedures reported in this study that were performed by NCI-CCR affiliated staff were approved by the NCI Animal Care and Use Committee (ACUC) and in accordance with federal regulatory requirements and standards. All components of the intramural NIH ACU program are accredited by AAALAC International.

### Oligos, plasmids, and mRNA synthesis
The cDNA clone that encoded full-length *Xenopus Ror2* was obtained from Source BioScience (GenBank ID: BC169600). Various deletion mutants and ephrinB2-Y2F/E mutants were generated in the pCS107 vector using the QuikChange II Site-Directed Mutagenesis Kit. We labeled our constructs based on their tag location. For example, "ephrinB2-HA" has a C-terminal HA tag, "HA-ephrinB2" has an N-terminal HA tag. Capped sense RNAs were transcribed using the mMessage mMachine SP6 kit (ThermoFisher) after linearization with ASP718 or ApaI. Oligos information in Supplementary Data 2.

### Identification of ephrinB2-binding partners using IP/mass spectrometry analysis
In all, 100 pg of ephrinB2-HA RNAs were injected into two D1.1 blastomeres at 8 cell stage embryos. Neural plate explants were dissected from 14 to 16 stage embryos and analyzed by LC/MS/MS after trypsin/LysC digestion. The LC/MS/MS analysis of samples were carried out using a Thermo Scientific Q-Exactive hybrid Quadrupole-Orbitrap Mass Spectrometer and a Thermo Dionex UltiMate 3000 RSLCnano System. Raw data file acquired from each sample was searched against Xenopus protein sequences using the Proteome Discoverer 1.4 software (Thermo, San Jose, CA) based on the SEQUEST algorithm. Based on the PSM # (peptide spectrum match counts, an indicator of abundance), the relative abundance of a protein can be determined for comparison between two samples.

### Hybridization chain reaction RNA fluorescence in situ hybridization (HCR RNA-FISH)
HCR probe sets were designed by Molecular Instruments. *Xenopus* embryos were collected at stage 15 and then processed for HCR following the HCR v3.0 protocol for whole-mount *zebrafish* embryos. For the cross-section, Embryos were embedded in 4% low-melting agarose gel (2070-OP, Sigma) and were sectioned with a thickness of 100 μm with the vibratome (LEICA VT 1200 S). HCR images were taken by confocal microscopy (Zeiss LSM880).

## Immunofluorescence and confocal microscopy

Ectodermal explants (animal caps) or neural plate explants were dissected from stage 10.5 or 15 embryos and immunofluorescence microscopy was carried out using standard protocols. Briefly, the explants were fixed within MEMFA (4% formaldehyde in 1× MEM salt) at 4 °C for overnight or 2% Trichloroacetic acid (TCA) for 30 min at room temperature and then dehydrated with 100% methanol. The following primary antibodies were incubated after blocking with the filtered 10% goat serum in 1x PBS: Rabbit anti-ephrinBs (1:1,000, 600-401-MP0, Rockland), Mouse anti-Ror2 (1:50, Ror2, DSHB), Mouse anti-C-cadherin (1:50, 6B6, DSHB), Mouse anti-ZO1 (1:250, ZO1-1A12, ThermoFisher), Rabbit anti-pMLC (1:250, ab2480, Abcam), Mouse anti-HA-Alexa Fluor-488/555/647 (1:500, 2-2.2.14, ThermoFisher), Mouse anti-V5-Alexa Fluor-488/555/647 (1:500, 2F11F7, ThermoFisher), and Rabbit anti-GFP-Alexa Fluor-488 (1:500, A-21311, ThermoFisher). The secondary antibodies used were Alexa Fluor-488 or Alexa Fluor-594 conjugated Goat anti-rabbit IgG or anti-mouse IgG (1:500, Invitrogen). For F-actin staining, Alexa Fluor™ 488 Phalloidin (A12379, ThermoFisher). The samples were mounted and imaged using Zeiss LSM880 with Airyscan laser scanning confocal microscope.

## Live-cell imaging with F-actin probe (GFP-UtrCH)

For live imaging, we adopted a protocol from a previous study[89]. MOs and RNAs were injected along with 50 pg RNA of GFP-UtrCH into the D1.1 blastomere at the 16-cell stage. Early neurula stage embryos (stage 14–15) were placed in glass-bottom dishes containing 0.1x MBS. Embryos were arranged to have the neural plate facing down in the dish by adding high vacuum grease and agarose pieces and then gently laid down the cover slip over the embryos to make the neural plate flat. Maintaining the proper focus on dynamic moving tissues during neural tube closure was technically difficult under our live-cell confocal microscopy set-up. We were only able to maintain the live tissue imaging for less than 10 min. Time-lap images were taken from a single plane near the apical side of neural plate cells using continuous scanning (15 seconds per frame) on the Zeiss LSM880.

## Co-immunoprecipitation and western blot analysis

Embryos were lysed in lysis buffer (50 mM Tris-HCl [pH 7.4], 150 mM NaCl, 1% NP-40, 0.5 mM phenylmethyl sulphonyl fluoride (PMSF, ThermoFisher), protease inhibitor cocktail (Roche), and phosphatase inhibitors (100 mM Sodium Vanadate and 10 mM β-glycerophosphate). The cell lysates were sonicated and cleared by centrifugation at $13,000 \times g$ for 10 min at 4 °C. IPs were performed at 4 °C for 8 h with the following agarose beads: Anti-HA-agarose (Sigma-Aldrich), Anti-Flag-agarose (Sigma-Aldrich), Anti-myc-agarose (Sigma-Aldrich), GFP-Trap affinity resin (Chromotek), and GFP-V5 affinity resin (Chromotek). Western blot analysis was performed using anti-Flag-HRP-conjugated (1:5000, Sigma-Aldrich), anti-HA-HRP-conjugated (1:5000, Sigma-Aldrich), anti-myc-HRP-conjugated (1:5000, Sigma-Aldrich), anti-GFP-HRP-conjugated (1:5000, Rockland Immunochemicals), anti-V5-HRP-conjugated (1:5000, ThermoFisher), anti-ERK2 (1:1000, Santa Cruz Biotechnology), anti-b-actin (1:1000, Santa Cruz Biotechnology) and mouse anti-alpha-tubulin-HRP-conjugated (Proteintech). Secondary antibodies used were goat anti-rabbit-HRP-conjugated (Cell Signaling Technology) and mouse anti-goat-HRP-conjugated (Santa Cruz Biotechnology). HRP-dependent luminescence was developed using ECL Western Blotting Substrate (Pierce Biotechnology) and detected using X-ray film (Agfa HealthCare) or a ImageQuant™ LAS 4000 (GE Healthcare Life Science).

## Knockout using CRISPR/Cas9

For guide RNA design, the CRISPRscan website (https://www.crisprscan.org) was used to scan the *Xenopus laevis* genome for suitable Cas9 target sequences. The 5′ oligonucleotides (5′- CTAGCTAATA CGACTCACTATA-20 nt GTTTTAGAGCTAGAAATA-3′) containing each specific sgRNA sequence (20 nucleotides) and the 3′ common oligonucleotides (5′-AAAAGCACCGACTCGGTGCCACTTTTTCAAGTTGA-TAACGGACTAGCCTTATTTTAACTTGCTATTTCTAGCTCTAAAAC-3′) were synthesized and then amplified by PCR for sgRNA templates. sgRNAs were transcribed using the MEGAscript™ T7 Transcription Kit (ThermoFisher). Single-guide RNA (sgRNA) template construction, in vitro transcription of sgRNA, microinjection and genotyping were performed as described in Nakayama et al.[90]. For CRISPR/Cas9-mediated knockout, 200 pg ephrinB2, Ror2, or Wnt4 sgRNA along with 1 ng Cas9 protein (PNA Bio Inc.) was injected at the one-cell stage. F0 embryos were analyzed after imaging to verify genomic editing by the direct sequencing of PCR amplicons assay.

## Whole-mount in situ hybridization (WISH)

*Xenopus* embryos were collected at stage 16 or 20 for hybridization with the *Sox2* or *Wnt4* probes. Embryos were injected with membrane-GFP and various mRNAs or MOs to distinguish the injected side of embryos. The embryos were then processed for whole-mount in situ hybridization using standard methods[91].

## Reverse transcription and quantitative real-time PCR

Reverse transcription was carried out by using a SuperScript™ IV First-Strand Synthesis System (ThermoFisher). The PCR reactions were performed with SYBR™ Green PCR Master Mix (4309155, Applied Biosystems™) using QuantStudio™ 5 Real-Time PCR System (A34322, Applied Biosystems™). *ODC* was used for normalization.

## Statistics and reproducibility

The pilot experiments were conducted for all experiments to evaluate RNA and MO dosage and then full-scale experiments were performed at least three independent times. All experiments were performed blinded with an order of testing randomized. The sample size was determined as indicated in the figures and the specific statistical method was not used. Dead cells and embryos were excluded from all experimental analysis. Embryos that have mistargeted injection also were excluded before analysis (the target injection was confirmed by co-injection of lineage tracer (membrane-GFP/RFP RNAs). ImageJ program was used for all quantification. Normality of data was tested using Kolmogorov–Smirnov's test, D'Agostino and Pearson omnibus normality test and Shapiro–Wilk normality test using Prism 9 (version: 9.4.1). The data were considered as normal if found as normal by all three tests. Data sets following a normal distribution was compared with Student's t-test (two-tailed, unequal variances) or a one-way analysis of variance (ANOVA) with Dunnett's multiple comparisons post-test in Prism9. The data that did not follow a normal distribution were compared using Mann–Whitney's test or a non-parametric ANOVA (Kruskal–Wallis with Dunn's multiple comparisons post-test) using Prism 9. Cross comparisons were performed only if the overall $p$-value of the ANOVA was <0.05.; error bars: SEM.

## Reporting summary

Further information on research design is available in the Nature Portfolio Reporting Summary linked to this article.

## Data availability

Raw mass spectrometry Data are supplied as Supplementary Data 1 file. All source data presented in graphs within the Figures and uncropped scans of all blots and gels in Figures are provided as Source Data files. Additional information is available from the corresponding authors J.Y. (jaeho.yoon@nih.gov) upon request. Source data are provided with this paper.

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

## Acknowledgements

We thank Ming Zhou for performing the Mass spectrometric analysis of our ephrinB2 immunoprecipitants. We thank the following for providing DNA constructs: myc-Shroom3 and Wnt ligands from Sergei Sokol, GFP-Tricellulin from Ann Miller, and Flag-Rock from Jin-Kwan Han. This research was supported by the Intramural Research Program of the National Institutes of Health, National Cancer Institute.

## Author contributions

J.Y. performed all the experiments with the help of J.S., M.L., and Y.H. J.Y. and I.O.D. designed experiments, interpreted data, and wrote the manuscript. I.O.D. supervised the project. All authors discussed the results and reviewed the manuscript.

## Funding

## Competing interests

The authors declare no competing interests.
