## [Peer Review File · Nature Communications]

Wnt4 and ephrinB2 instruct apical constriction via Dishevelled and non-canonical signalingREVIEWER COMMENTS

Reviewer #1 (Remarks to the Author):

This manuscript reports a novel signaling complex comprising Wnt4, Ror2, Dsh2, EphrinB2 and Shroom3 (abbreviated WERDS), which is required for neural tube closure in *Xenopus* embryos. The WERDS complex mediates Wnt non-canonical signaling and promotes apical constriction in neural plate cells through actomyosin contraction. Neural tube closure is an important developmental process but its molecular regulation is not fully understood. The pathway uncovered is novel, and the involvement of the different proteins is carefully documented through multiple approaches and high-quality experiments with appropriate controls. Thus, the significance of the work is high, and the findings are of general interest.

There are some minor concerns that could be addressed to further improve the manuscript.

It is not clear what regulates the specificity of Wnt4 interaction with ephrinB2 (as shown in Fig. 3a). Does it involve specific binding of Wnt4 to Ror2 in the context of the WERDS complex? Could the authors address in the text how they envision the specificity could arise?

Page 21. In the "plasmids" section, more information should be included on the constructs used. For example, the location of the tags should be indicated. Are the HA and V5 tags at the C-terminus of ephrinB2? If so, is the PDZ binding motif inactivated by the addition of the tag? (Since PDZ domain binding motifs typically must be at the C-terminus to be functional) Is Dsh2 also tagged at the C-terminus? If so, could the presence of the tag weaken the intramolecular interaction of the Dsh2 PDZ domain-binding motif with the Dsh2 PDZ domain and make it more susceptible to inhibition by ephrinB2?

The manuscript shows that the interaction between Dsh2 and ephrinB2 requires a few C-terminal amino acids in each protein. Can the authors briefly discuss how they envision this interaction works? Are the identified C-terminal amino acids of Dsh2 and ephrinB2 hypothesized to be solely responsible for the interaction? Or to be required but possibly function in concert with other regions? Is the interaction between ephrinB2 and Dsh2 independent of the PDZ domain binding ability of the motifs? Is the interaction likely to be direct? It would be useful to have a more clear idea of the limitations of the study and what questions still need to be answered in future work.

Very minor concerns

Please add a brief explanation about what the GFP-UtrCH construct encodes and how it works.

Page 12. Briefly explain what "tricellular junctions" means.

Fig. 4b. Should "PBD" be "PBM", to be consistent with the rest of the manuscript?

Figs. 4e and S3g. What does deltaCNR mean? (Is CNR the same as SRP?)

Fig. 7a. There appears to be a typo following the ephrinB2 name near the third and fifth row of the IF images.

Fig. S3d. In the legend on top for Dsh2, should the second and third "+" be removed?

Fig. 6f. The name "Shroom3" should be included next to the domain structure; should "PDZ" in Shroom3 be "SRP"? Why are the colors of the Dsh2 C-terminal part different in the right-most panel compared to the other panels?

Do the authors know why in some blots ephrinB2 mutants with C-terminal deletions or the ephrinB2 Y2F mutant have an apparent molecular weight substantially higher than ephrinB2 WT? (Figs. 1f, 4f, 7b).

Reviewer #2 (Remarks to the Author):

The manuscript by Yoon et al describes an important and partially novel signaling complex that is required for neural tube morphogenesis. Overall, the majority of the data presented is of exceptional quality and presents a very compelling story that is of high significance in the field of cell and developmental biology. However, I have concerns that some of the conclusions are based on presumptive leaps and not direct experimental evidence presented with the manuscript. I also feel there are some deficits in describing the methods used in some aspects of the analysis. Concerns are the following:

1. In Figure 1h and 1i: the graphs are depicting “% of apical constriction defects” on the Y-axis. However, I was unable to determine in the text where or how the authors measured apical constriction in the experiments depicted. Is it being assumed that the observed defects in NT closure in the morphants is the result of defects in apical constriction or was it determined by measuring apical area of cells or apical width (as in figure 2)? The authors should clarify this, change the graphs to reflect what was actually used to determine the effect of the morpholinos, or provide the data showing defects in apical constriction.

2. In Figure 2:

a. I am assuming that the apical width measurement is for individual cells and the distance from the medial to lateral position in the figures shown?

b. It is important to know where and how actin intensity was determined; line segments across the junctions, intensity of an ROI, etc. Also, was actin intensity normalized to another marker, such as memRFP? This might be important since if cells are apically constricted (or not) this could influence fluorescent intensity readings.

c. In figure 2C, is the major difference in GFP-UtrCH distribution in the apical-medial actin? If so, this might provide interesting insights in the types of actin networks required/regulated by the WERDS complex.

d. Finally, the authors state the WERDS complex regulates “contractile actin network formation”. While this conclusion is implied from the data and videos presented, I do not think it was sufficiently documented that these networks are indeed contractile or the process is myosin-dependent. I think some of the videos already presented could be used to quantify contractility.

3. There are some important references that should be included:

a. A link between Shroom3 and Dvl2 in PCP signaling in vivo has also been shown in the paper by Durbin et al, 2020 (DOI: 10.1016/j.ydbio.2020.05.013).

b. The importance of SHROOM3 in human NTD has been demonstrated in Deshwar et al, 2020 (<https://doi.org/10.1111/cge.13804>). In which they describe a novel recessive lethal human SHROOM3 mutation causing neural tube defects that mimic the mouse null mutations that have been published.

4. Following up with point 1 above, I was unable to easily verify in the cited references that EphrinB2-Y2F and Y2E are able and unable, respectively, to interact with Dsh2 (pages 15-16). One paper seems to focus on EphrinB1 and the other does not seem to have those specific mutations in EphrinB2 (at least that I could determine in the figures). This either needs to be clarified specifically in text or these data need to be included in the manuscript.

5. Since the authors make a significant point about the regulation of the Dsh-Ephrin interaction being modulated by tyrosine phosphorylation, is there any indication that Ror2 kinase activity is required or participates in forming/disassembling the WERDS complex? Similarly is there any indication that Ror2 may be activated by Wnt4, as it seems to be by Wnt5a? I think this possibility might be worthy of discussion.

6. Finally, based on the model proposed regarding the closed vs. open conformation of Dsh, would it be useful to test if a Dsh variant with a point mutation in the PDZ domain is more efficient at assembling the WERDS complex? These data could help support the idea that this is an intramolecular

interaction between the PDZ and c-terminus. It could also be an intermolecular interaction between multimers of Dsh proteins. I appreciate that results from this experiment could be difficult to interpret based on the fact that the PDZ may mediate interactions with other proteins. At a minimum it might also be useful to address the concept that other interactions with PDZ domain may facilitate formation of the open conformation.

Reviewer #3 (Remarks to the Author):

The manuscript by Yoon et al. describes comprehensive sets of experiments that address signal crosstalk between Wnt4/Ror2/Dsh2 PCP and EphrinB2 in regulating Shroom3 and ROCK activation in apical constriction of neural cells during neural tube closure (NTC) in *Xenopus*. Though many of the individual components have been shown previously to be important for NTC, how these factors act together, independently, or sequentially is not known. The current study reveals a hierarchical multiprotein complex that links different signals and actin regulators to influence F-actin distribution and apical constriction in NTC. The experiments were thorough, including rescue with wild type or mutant constructs for all the MO studies. CRISPR/Cas9 phenotypes were provided alongside with the MO-injected embryos. Reciprocal co-IPs to test protein interactions were performed. Quantification and statistics were considered carefully for all the relevant experiments. The data were of high quality and generated the insight that EphrinB2 crosstalks with the PCP pathway via regulation of Dsh2 conformation, resulting simultaneously in inhibition of canonical Wnt signal and enhancement of a PCP/Shroom3 complex that facilitates actin-myosin networks for apical constriction.

The work is generally appropriate for publication in *Nature Communications*. However, some points need to be clarified prior to publication.

The data on inhibition of canonical Wnt signaling by EphrinB2, though solid, seem to be out of focus of the issues at hand. The study deals with a signaling complex (WERDS) involved in regulation of apical constriction and neural tube closure. It is unclear whether inhibition of canonical Wnt signaling is required for EphrinB2 to regulate NTC, or formation of WERDS is sufficient to induce apical constriction even in the presence of canonical Wnt signal. Since the data can distract the main theme, it is suggested that Figure 7 be removed or put in Supplementary section.

In Fig. 1i and 3c, quantification of neural tube defects was described as % apical constriction defects in the y-axis. This should be corrected, as the experiments did not address apical constriction per se.

It is described that Shroom3, Ror2, and EphrinB2 are localized at the proximal tricellular junctions in Fig. 5. A diagram for the orientation of the neural plate and the region of image should be provided in this figure, as the readers will not be able to distinguish proximal-distal or anterior-posterior axis from the images.

For live imaging of actin-bundle formation, additional information on experimental setup, such as z-stacks and the length of the movies, should be provided.

All the reviewers' assessments are much appreciated.

Reviewer #1 (Remarks to the Author):

This manuscript reports a novel signaling complex comprising Wnt4, Ror2, Dsh2, EphrinB2 and Shroom3 (abbreviated WERDS), which is required for neural tube closure in Xenopus embryos. The WERDS complex mediates Wnt non-canonical signaling and promotes apical constriction in neural plate cells through actomyosin contraction. Neural tube closure is an important developmental process, but its molecular regulation is not fully understood. The pathway uncovered is novel, and the involvement of the different proteins is carefully documented through multiple approaches and high-quality experiments with appropriate controls. Thus, the significance of the work is high, and the findings are of general interest. There are some minor concerns that could be addressed to further improve the manuscript.

We thank the reviewer for this assessment, and have addressed the minor comments below

It is not clear what regulates the specificity of Wnt4 interaction with ephrinB2 (as shown in Fig. 3a). Does it involve specific binding of Wnt4 to Ror2 in the context of the WERDS complex? Could the authors address in the text how they envision the specificity could arise?

The reviewer makes an interesting point. As shown in Fig 1B, the WERDS complex and Wnt5 showed distinct spatial expression patterns. Wnt4, Ror2, and ephrinB2 are expressed in neural plate cells but Wnt5 is expressed in pre-somatic mesodermal cells. This result suggests that specific spatial expression patterns may contribute to the specificity of Wnt4 interaction with ephrinB2. Interestingly, we also found using an IP assay with Wnt ligands overexpressed in embryos that Wnt4 is specifically found in the ephrinB2 immune-complexes. Furthermore, the interaction between ephrinB2 and Ror2 was elevated by Wnt4, but not by Wnt5 (a known Ror2 ligand). As the reviewer suggested, we examined whether Ror2 specifically binds with Wnt4 in developing *Xenopus* embryos. Our Co-IP results clearly showed that Ror2 interacts with both Wnt4 and Wnt5a (non-canonical Wnt ligands), but not Wnt8 (canonical Wnt ligand) (now presented as Supplementary Fig. S2d). However, we did not observe a significantly different binding affinity between Wnt4 and 5.

Previously, Sheetz et al (2020) suggests that the specific ligand binding can modulate the conformational changes resulting in interaction with specific binding partners. There is another possibility - that Wnt4 induces specific phosphorylation of Ror2 which modulates binding partner preferences. Further studies are needed to better understand what regulates the specificity of the Wnt4 interaction with ephrinB2. We have now added these data (Supplementary Fig. S2d) and discuss these possibilities in the results section.

Page 21. In the "plasmids" section, more information should be included on the constructs used. For example, the location of the tags should be indicated. Are the HA and V5 tags at the C-terminus of ephrinB2? If so, is the PDZ binding motif inactivated by the addition of the tag? (Since PDZ domain binding motifs typically must be at the C-terminus to be functional) Is Dsh2 also tagged at the C-

terminus? If so, could the presence of the tag weaken the intramolecular interaction of the Dsh2 PDZ domain-binding motif with the Dsh2 PDZ domain and make it more susceptible to inhibition by ephrinB2?

The reviewer makes an excellent point. We have revised the labeling of our constructs based on their tag location. For example, “*ephrinB2-HA*” has a C-terminal HA tag, “*HA-ephrinB2*” has an N-terminal HA tag. Next, to test whether the C-terminal tag affects the ephrinB2/Dsh2 interaction, we constructed N-terminal tagged ephrinB2 by inserting the HA tag after the signal peptide (SP). Co-IP using GFP-Dsh2 and N- or C-terminal tagged ephrinB2 reveals that the C-terminal tag does not hinder the interaction between Dsh2 and ephrinB2 (now presented as Supplementary Fig. S3e). We also checked whether the C-terminal tag disturbs the intramolecular interaction of Dsh2. We performed Co-IP analysis using GFP-Dsh2- Δ PBM (lacking the PDZ-Binding-Motif and therefore prevents intramolecular interaction) and N- or C-terminal tagged Dsh2 tail constructs. These Co-IPs demonstrated that the C-terminal small tag (HA, 9 amino acids) does not hinder the intramolecular interaction between the PDZ domain and C-terminal PBM (PDZ domain binding motif) (now displayed in Supplementary Fig. S3f).

The manuscript shows that the interaction between Dsh2 and ephrinB2 requires a few C-terminal amino acids in each protein. Can the authors briefly discuss how they envision this interaction works? Are the identified C-terminal amino acids of Dsh2 and ephrinB2 hypothesized to be solely responsible for the interaction? Or to be required but possibly function in concert with other regions? Is the interaction between ephrinB2 and Dsh2 independent of the PDZ domain binding ability of the motifs? Is the interaction likely to be direct? It would be useful to have a more clear idea of the limitations of the study and what questions still need to be answered in future work.

The reviewer asks a very interesting question, and the short answer is that we don't know. We currently can only say that these two C-terminal regions are necessary for the interaction. It is possible that there is a direct interaction between the C-termini of these proteins and these last amino acids are required for the interaction. It is also possible that there is, as yet, unidentified bridging protein between ephrinB2 and Dsh2, or as mentioned by the reviewer, that these motifs are necessary to allow proper association among other portions of the proteins. We now mention these possibilities and limitation in our Discussion section.

Very minor concerns

Please add a brief explanation about what the GFP-UtrCH construct encodes and how it works.

As suggested, we added a brief explanation of the GFP-UtrCH fusion protein. UtrCH is the calponin homology domain of utrophin, which binds F-actin without stabilizing it in living cells and tissues (Burkel *et al*, 2007).

Page 12. Briefly explain what “tricellular junctions” means.

As suggested, we added a brief explanation of the tricellular junctions. In vertebrate epithelial cells, a

specialized structure, the tricellular tight junction, has been identified at the place where three cells meet (Higashi *et al*, 2017).

Fig. 4b. Should “PBD” be “PBM”, to be consistent with the rest of the manuscript?

We have now corrected this error.

Figs. 4e and S3g. What does deltaCNR mean? (Is CNR the same as SRP?)

It should be SRP. we have made the correction.

Fig. 7a. There appears to be a typo following the ephrinB2 name near the third and fifth row of the IF images.

It has been corrected.

Fig. S3d. In the legend on top for Dsh2, should the second and third “+” be removed?

We have now corrected this error.

Fig. 6f. The name “Shroom3” should be included next to the domain structure; should “PDZ” in Shroom3 be “SRP”? Why are the colors of the Dsh2 C-terminal part different in the right-most panel compared to the other panels?

We are sorry for the confusion. “PDZ” in Shroom3 should be “SRP”. The different colors of Dsh2 C-terminal portion were an error. We have made the correction.

Do the authors know why in some blots ephrinB2 mutants with C-terminal deletions or the ephrinB2 Y2F mutant have an apparent molecular weight substantially higher than ephrinB2 WT? (Figs. 1f, 4f, 7b).

This is an interesting property of both ephrinB1 and ephrinB2 (probably due to their high homology in the cytoplasmic domain) that has been noticed by us and several colleagues over the years. We believe that the C-terminal deletion or Y2E mutation alters the protein structure and perhaps charge resulting in the electrophoretic mobility shift observed in SDS gel electrophoresis.

Reviewer #2 (Remarks to the Author):

The manuscript by Yoon et al describes an important and partially novel signaling complex that is required for neural tube morphogenesis. Overall, the majority of the data presented is of exception quality and presents a very compelling story that is of high significance in the field of cell and developmental biology. However, I have concerns that some of the conclusions are based on presumptive

leaps and not direct experimental evidence presented with the manuscript. I also feel there are some deficits in describing the methods used in some aspects of the analysis. Concerns are the following:

We appreciate the reviewer's assessment and have addressed the concerns below.

1. In Figure 1h and 1i: the graphs are depicting “% of apical constriction defects” on the Y-axis. However, I was unable to determine in the text where or how the authors measured apical constriction in the experiments depicted. Is it being assumed that the observed defects in NT closure in the morphants is the result of defects in apical constriction or was it determined by measuring apical area of cells or apical width (as in figure 2)? The authors should clarify this, change the graphs to reflect what was actually used to determine the effect of the morpholinos, or provide the data showing defects in apical constriction.

The reviewer makes an excellent point. We actually measured neural tube closure defects. We have now corrected this error. The graphs in Figure 1h and 1i display “% of neural tube closure defects” in the current Figures.

2. In Figure 2:

a. I am assuming that the apical width measurement is for individual cells and the distance from the medial to lateral position in the figures shown?

The reviewer is correct. We measured the apical width of individual cells from medial to lateral position of morpholino injected neural plates. We have now added a scheme (Fig. 2a) depicting how we measured the apical width.

b. it is important to know where and how actin intensity was determined; line segments across the junctions, intensity of an ROI, etc. Also, was actin intensity normalized to another marker, such as memRFP? This might be important since if cells are apically constricted (or not) this could influence fluorescent intensity readings.

The reviewer makes an excellent point. During neural tube closure, apical actin and phospho-MLC levels have dynamic changes during the timeline of development. As the reviewer mentioned, apically constricted cells showed much higher fluorescent intensity compared to less constricted cells. Neural plate cells start to present p-MLC and have similar apical area sizes at stage 15, as shown in supplementary Fig. S2. Therefore, we analyzed apical actin and p-MLC levels at stage 15. Despite harvesting embryos at the same time, significant variation in apical actin and p-MLC levels was observed among neural plate cells. For a more accurate analysis, Morpholino antisense oligonucleotides (MOs) were injected into the D.1.1 blastomere. These unilateral injections were performed such that the uninjected side served as an internal control. We measured apical actin and p-MLC intensity of an ROI (Region of interest) from uninjected and MO injected neural plate cells in one embryo and then normalized the actin intensity using an uninjected neural plate (internal control).

The histogram depicts the relative level of apical actin and p-MLC compared to the uninjected internal control. We have added detailed information in the revised manuscript on how we measured apical actin and p-MLC level.

c. In figure 2C, is the major difference in GFP-UtrCH distribution in the apical-medial actin? If so, this might provide interesting insights in the types of actin networks required/regulated by the WERDS complex.

We believe that WERDS complex regulates apical-medial actin bundle formation. Shroom3 has been proposed as a critical player for modulating apical-medial actin accumulation during neural tube closure of the *Xenopus* embryo (Baldwin *et al*, 2022 elife). Furthermore, the apical-medial actin formation in our live imaging results showed a very similar contractile medial-apical actomyosin network that had previously been presented (Arnold *et al*, 2019 elife).

d. Finally, the authors state the WERDS complex regulates “contractile actin network formation”. While this conclusion is implied from the data and videos presented, I do not think it was sufficiently documented that these networks are indeed contractile or the process is myosin-dependent. I think some of the videos already presented could be used to quantify contractility.

The reviewer’s point is well-taken. While our videos (movie 3, 4, and 6) clearly showed that the WERDS complex regulates medial-apical actin formation, we found it technically difficult to quantify contractility. Maintaining the proper focus on dynamic moving tissues during neural tube closure was technically difficult under our live cell confocal microscopy set-up. We were only able to maintain the live imaging for less than 10 min, which we believe is insufficient for accurately measuring the contractility. However, our results clearly showed that the WERDS complex regulated apical contractility (Fig. 2a and 3d) and phospho-MLC levels (supplementary Fig. S2g).

3. There are some important references that should be included:

a. A link between Shroom3 and Dvl2 in PCP signaling in vivo has also been shown in the paper by Durbin et al, 2020 (DOI: 10.1016/j.ydbio.2020.05.013).

b. The importance of SHROOM3 in human NTD has been demonstrated in Deshwar et al, 2020 (<https://doi.org/10.1111/cqg.13804>). In which they describe a novel recessive lethal human SHROOM3 mutation causing neural tube defects that mimic the mouse null mutations that have been published.

We thank the reviewer for the suggestions. We have now included the references.

4. Following up with point 1 above, I was unable to easily verify in the cited references that EphrinB2-Y2F and Y2E are able and unable, respectively, to interact with Dsh2 (pages 15-16). One paper seems to focus

on EphrinB1 and the other does not seem to have those specific mutations in EphrinB2 (at least that I could determine in the figures). This either needs to be clarified specifically in text or these data need to be included in the manuscript.

The reviewer makes an excellent point. Previously, we demonstrated that phosphorylation of tyrosines 324 and 325 in ephrinB1 disrupts the ephrinB1/Dsh interaction, (Lee et al, 2009 Mol Biol Cell.). Since ephrinB1 and ephrinB2 have a highly conserved intracellular domain, we tested whether phosphorylation of tyrosines in the PDZ binding motif (PBM) of ephrinB2 also affects the ephrinB2/Dsh interaction. Consistent with the ephrinB1/Dsh2 interaction, Co-IP analysis confirmed that ephrinB2 Y2E mutant (phosphomimetic mutant) did not interact with Dsh2 but the Y2F mutant (non-phosphorylatable mutant) strongly interacts with Dsh2. This data is now included as Supplementary Fig. S4b.

5. Since the authors make a significant point about the regulation of the Dsh-Ephrin interaction being modulated by tyrosine phosphorylation, is there any indication that Ror2 kinase activity is required or participates in forming/disassembling the WERDS complex? Similarly is there any indication that Ror2 may be activated by Wnt4, as it seems to be by Wnt5a? I think this possibility might be worthy of discussion.

We agree with the reviewer that this issue should be addressed; Ror2 has been reported to be a receptor tyrosine kinase (RTK), although recent studies indicate it is actually a pseudo-kinase (Sheetz et al, 2020 Mol. Cell). Since ephrinB2 has been shown to be phosphorylated on tyrosine residues within the intracellular domain, which modulates certain downstream signaling events, we tested whether Wnt ligands can induce the tyrosine phosphorylation of ephrinB2 via endogenous Ror2. Six different tagged Wnt ligands representing canonical and non-canonical Wnts were overexpressed along with ephrinB2 in embryos and Western blot analysis was performed on ephrinB2 immune complexes using specific antibodies for ephrinB2 tyrosine phosphorylation. Constitutively active FGF receptor, a positive control, induced phosphorylation at three tyrosine sites. However, none of the Wnt ligands induced tyrosine phosphorylation of ephrinB2 (now presented as Supplementary Fig. 4e). In addition, we also confirmed that overexpressing Ror2 along the Wnt4 ligand did not induce tyrosine phosphorylation of ephrinB2 (now presented as Supplementary Fig. 4f). We have included these results in the revised manuscript.

6. Finally, based on the model proposed regarding the closed vs. open conformation of Dsh, would it be useful to test if a Dsh variant with a point mutation in the PDZ domain is more efficient at assembling the WERDS complex? These data could help support the idea that this is an intramolecular interaction between the PDZ and c-terminus. It could also be an intermolecular interaction between multimers of Dsh proteins. I appreciate that results from this experiment could be difficult to interpret based on the fact that the PDZ may mediate interactions with other proteins. At a minimum it might also be useful to address the concept that other interactions with PDZ domain may facilitate formation of the open conformation.

We thank the reviewer for this very interesting suggestion. Jurásek and his colleagues demonstrated that Dsh3 also has an interaction between the C-terminal PBM (PDZ binding motif) and its own PDZ

domain (Jurásek *et al*, 2021 Sci Rep). This interaction can be regulated by post-translational modification including phosphorylation. Interestingly, the point mutation (Serine 263 to Glutamic acid) suppressed the intramolecular interaction of the PDZ domain and C-terminal motif resulting in the open conformation of Dsh. Since Dsh2 and 3 have a highly conserved PDZ domain and PBM, we made the same single amino acid mutation in Dsh2-S267E to examine whether the open conformation of Dsh2 is more efficient at interacting with WERDS components. Co-IP analysis showed that the Dsh2-S267E mutant has a more robust interaction with Ror2 and ephrinB2 when compared to wild-type (now displayed in Supplementary Fig. S3j and k). This result supports our model that ephrinB2 binds Dsh2 resulting in the open conformation of Dsh2 and allowing an enhanced Ror2 interaction.

Reviewer #3 (Remarks to the Author):

The manuscript by Yoon et al. describes comprehensive sets of experiments that address signal crosstalk between Wnt4/Ror2/Dsh2 PCP and EphrinB2 in regulating Shroom3 and ROCK activation in apical constriction of neural cells during neural tube closure (NTC) in Xenopus. Though many of the individual components have been shown previously to be important for NTC, how these factors act together, independently, or sequentially is not known. The current study reveals a hierarchical multiprotein complex that links different signals and actin regulators to influence F-actin distribution and apical constriction in NTC. The experiments were thorough, including rescue with wild type or mutant constructs for all the MO studies. CRISPR/Cas9 phenotypes were provided alongside with the MO-injected embryos. Reciprocal co-IPs to test protein interactions were performed. Quantification and statistics were considered carefully for all the relevant experiments. The data were of high quality and generated the insight that EphrinB2 crosstalks with the PCP pathway via regulation of Dsh2 conformation, resulting simultaneously in inhibition of canonical Wnt signal and enhancement of a PCP/Shroom3 complex that facilitates actin-myosin networks for apical constriction. The work is generally appropriate for publication in Nature Communications. However, some points need to be clarified prior to publication.

We thank the reviewer for this assessment and have addressed the minor comments below.

The data on inhibition of canonical Wnt signaling by EphrinB2, though solid, seem to be out of focus of the issues at hand. The study deals with a signaling complex (WERDS) involved in regulation of apical constriction and neural tube closure. It is unclear whether inhibition of canonical Wnt signaling is required for EphrinB2 to regulate NTC, or formation of WERDS is sufficient to induce apical constriction even in the presence of canonical Wnt signal. Since the data can distract the main theme, it is suggested that Figure 7 be removed or put in Supplementary section.

We agree with the reviewer's viewpoint, although we feel the data is of enough general interest to retain in the manuscript, albeit less prominently. Thus, we have modified the manuscript as follows: 1) we have removed statements regarding antagonizing Wnt signaling in the abstract; 2) we moved figure

7 into the Supplementary section. Figure 7 supports the concept that ephrinB2 induces a conformational change in Dsh2, but as the reviewer notes - it is still unclear whether ephrinB2's role in antagonizing canonical Wnt signaling is a requirement for promoting neural tube closure. The move to the Supplementary section (Supplementary Fig. S4) avoids this distraction from our main theme.

In Fig. 1i and 3c, quantification of neural tube defects was described as % apical constriction defects in the y-axis. This should be corrected, as the experiments did not address apical constriction per se.

The reviewer is correct that the original Fig. 1i and 3c was mislabeled. We have now corrected this error both in the figures and the text to “% neural tube closure defects”.

It is described that Shroom3, Ror2, and EphrinB2 are localized at the proximal tricellular junctions in Fig. 5. A diagram for the orientation of the neural plate and the region of image should be provided in this figure, as the readers will not be able to distinguish proximal-distal or anterior-posterior axis from the images.

As suggested, we have now added diagrams for the orientation of neural plates.

For live imaging of actin-bundle formation, additional information on experimental setup, such as z-stacks and the length the movies, should be provided.

As suggested, we have added the additional information in “Methods and Materials section”.

REVIEWERS' COMMENTS

Reviewer #1 (Remarks to the Author):

The authors have fully addressed my minor concerns.

When reading, I noticed a couple of typos:

Page 7, second paragraph. Supplementary Fig. 1g should be Supplementary Fig. 1e and Supplementary Fig. 1h should be Supplementary Fig. 1f).

Fig. S3d. Should "Dsh2-deltaC9" be "Dsh2-deltaT9"?

Fig. 6f. "Cloe" should be "Close".

Reviewer #2 (Remarks to the Author):

I thank the authors for their thorough and comprehensive responses to my previous review; all of my concerns/comments have been addressed.

Reviewer #3 (Remarks to the Author):

The authors have addressed my concerns satisfactorily.

REVIEWERS' COMMENTS

Reviewer #1 (Remarks to the Author):

The authors have fully addressed my minor concerns.

We thank the reviewer for this assessment.

When reading, I noticed a couple of typos:

Page 7, second paragraph. Supplementary Fig. 1g should be Supplementary Fig. 1e and Supplementary Fig. 1h should be Supplementary Fig. 1f).

We have now corrected this error.

Fig. S3d. Should “Dsh2-deltaC9” be “Dsh2-deltaT9”?

The reviewer is correct. It has been corrected.

Fig. 6f. “Cloe” should be “Close”.

we have made the correction.

Reviewer #2 (Remarks to the Author):

I thank the authors for their thorough and comprehensive responses to my previous review; all of my concerns/comments have been addressed.

We thank the reviewer for this assessment.

Reviewer #3 (Remarks to the Author):

The authors have addressed my concerns satisfactorily.

We appreciate the reviewer's assessment.